# GraphPlan: Graph-enhanced Planning via Thinking LLMs for Embodied Agents

## Abstract

Embodied agents that follow instructions to complete complex tasks in visual environments have attracted increasing attention. Large Language Models (LLMs) based planners, notwithstanding the progress achieved, still suffer from three main limitations: (i) a lack of physical grounding, often resulting in hallucinatory plans; (ii) poor generalization to unseen long-horizon tasks; and (iii) an absence of environmental awareness in the open-loop planning process. To address these issues, we propose *GraphPlan*, a novel framework that integrates a *task graph* to provide structured knowledge for robust planning and a *scene graph* to maintain environmental memory for event-driven replanning. Specifically, the task graph guides the LLM's reasoning through contextual prompting and iterative refinement, effectively mitigating planning hallucinations. Furthermore, within the GRPO framework, the task graph offers delicate reward design to train LLMs' reasoning, enhancing long-horizon planning capabilities and improving generalization. Finally, the memory constructed by a dynamic scene graph empowers an event-driven replanning module, enabling the agent to foster environment awareness and correct instruction misalignment within a closed-loop planning process. On the benchmark ALFRED, GraphPlan achieves state-of-the-art performance on the official leaderboard. Moreover, its high-level planner outperforms a series of leading API-based LLMs on both the validation set and unseen long-horizon tasks. Additional experiments reveal the promising potential of our graph-enhanced framework in few-shot or zero-shot learning scenarios, and its generalization to novel tasks beyond the benchmark.

## 1 Introduction

There has been a growing exploration of embodied agents designed to execute long-horizon everyday tasks given human instructions (Zhang et al., 2024; Kim et al., 2024a;c; Cai et al., 2025). In the field of Embodied AI, instruction following necessitates that agents perform three key operations: interpreting natural language, using egocentric visual observations, and executing physical action to navigate and interact with the environment. A straightforward approach (Pashevich et al., 2021; Suganuma et al., 2021; Ehsani et al., 2024) involves training agents in an end-to-end supervised manner using large-scale datasets with annotated instructions and low-level expert action sequences. However, this paradigm is resource-intensive: it relies heavily on task-specific data and shows poor generalization to unseen scenarios. In contrast, data-efficient hierarchical methods (Min et al., 2021; Bhambri et al., 2023; Yang, 2024; Kim et al., 2025) have emerged as a promising alternative: the high-level planner first decomposes instructions into subtasks, while the low-level executor subsequently accesses the skill library to translate these subtasks into executable actions in the environment.

Recently, a growing trend of hierarchical methods involves Large Language Models (LLMs) harnessing for high-level planning (Ahn et al., 2022; Singh et al., 2023; Rana et al., 2023; Sun et al., 2025), thanks to the powerful language comprehension and reasoning capabilities of LLMs. When initial planning fails, these models can engage in replanning by incorporating observed environmental objects as contextual prompts (Song et al., 2023; Chen et al., 2025; Kim et al., 2025). Nonetheless, long-horizon planning remains a major challenge for embodied agents, with three key limitations: (i) Despite their strong reasoning capabilities, general-purpose LLMs lack physical grounding, leading to instruction misinterpretation, planning hallucinations, and increased failure rates as task

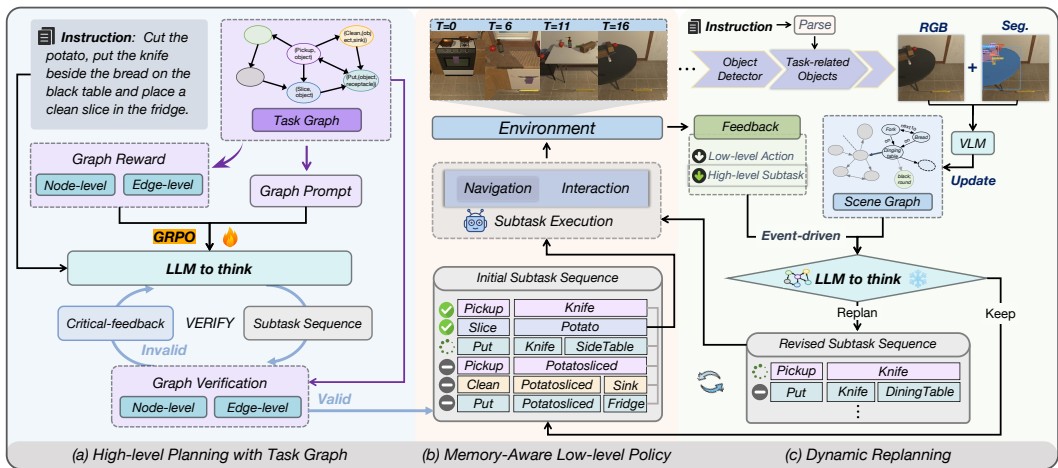

Figure 1: GraphPlan consists of three core modules: (a) the high-level planner with task graph generates an initial plan, (b) the memory-aware low-level policy executes navigation and interaction actions for each subtask, and (c) dynamic replanning adjusts plans during execution. The example here shows replanning triggered upon the completion of subtask "put(knife, sidetable)", and the revised subtasks become "pickup(knife)" and "put(knife, dining table)".

complexity grows. (ii) Supervised fine-tuning on limited in-domain data results in poor generalization to unseen long-horizon tasks, except when employing prohibitively expensive annotations. (iii) Existing replanning strategies, which only trigger corrections for low-level failures, are prone to myopic decisions and inefficient retries as they lack the environmental awareness to detect plans that are executable by low-level actions but misaligned with the instruction.

To address these challenges, we propose *GraphPlan*, a dual-graph enhanced closed-loop planning system, as illustrated in Figure 1. This framework integrates two core structured representations: a *task graph* to provide structured knowledge for robust planning, and a *scene graph* to maintain environmental memory for event-driven replanning. The **task graph** plays three essential roles in high-level planning. First, it guides the LLM's plan generation by explicitly incorporating the task graph into the prompt. Second, it serves as an external verification to detect planning hallucinations and refine the subtask sequence iteratively. Finally, we adopt Group Relative Policy Optimization (GRPO) (Shao et al., 2024) with task graph-based reward to enhance the reasoning capabilities of LLMs. Since embodied planning admits multiple valid solutions for a single goal, reward design becomes particularly challenging. The task graph addresses this issue by encapsulating diverse feasible paths, enabling reward signals that accommodate multiple valid solutions. With this delicate reward design, we are able to facilitate the alignment between the high-level action space and instructions, leading to stronger reasoning and generalization capabilities.

To enhance the LLM's environmental awareness and correct instruction misalignment within a closed-loop planning process, we design an event-driven replanning module powered by a dynamic scene graph. The dynamically updated *scene graph* serves as a task-centric memory module to focus reasoning on task-relevant environmental cues. When replanning is triggered by either low-level execution errors or high-level subtask completion, the current scene graph memory and plan execution progress are provided to the thinking LLM to support efficient inference, tracking, and adaptation of the plan. To ensure the quality of replanning, all proposed revisions are constrained by the task graph to maintain feasibility. Together, these components form a closed-loop system that continuously aligns plan execution with user intent under environmental feedback.

We evaluate GraphPlan on ALFRED (Shridhar et al., 2020), a challenging benchmark for vision-language navigation and interaction. Our hierarchical planning framework **ranks first on the official ALFRED leaderboard**. Particularly, we evaluate the performance of the high-level planner and *contribute a long-horizon dataset* extending ALFRED with 17 complex task types to better assess long-horizon generalization. The experimental results show that GraphPlan outperforms a range of leading API-based LLMs on both validation set and unseen long-horizon benchmark, with additional experiments indicating its promising extensibility to few-shot and zero-shot learning. Furthermore, task graph extension experiments validate effective generalization beyond the benchmark.

## 2 RELATED WORK

**Task Planning for Embodied Agents.** Prior work (Shridhar et al., 2020; Pashevich et al., 2021; Suganuma et al., 2021; Singh et al., 2021) train agents end-to-end to directly generate low-level actions given language instructions, but their performance in long-horizon tasks remains limited. Recently, hierarchical or modular planning (Inoue & Ohashi, 2022; Shi et al., 2024; Kim et al., 2025) have proven effective by decomposing tasks into subtasks to bridge the gap between natural instructions and executable actions. In the early stage, template-based methods (Min et al., 2021; Yang, 2024) are limited to predefined tasks and struggle to generalize. To address this problem, LLMs are being explored as high-level planners, either through few-shot in-context prompting (Song et al., 2023; Kim et al., 2025) or by supervised training on specific datasets (Zhao et al., 2024; Chen et al., 2025). Some works (Huang et al., 2022; Song et al., 2023; Kim et al., 2024b; 2025) further introduce replanning mechanisms to adjust actions by accepting environmental feedback, triggering local corrections to immediate errors or predefined state differences. Hence, we propose an event-driven dynamic replanning mechanism to enhance both plan feasibility and instruction alignment.

**Complex Reasoning with LLMs.** To solve complex reasoning tasks, Chain-of-Thought (CoT) methods (Wei et al., 2022; Cheng et al., 2023; Obi et al., 2025) and various adaptations of CoT reasoning have been proposed, which prompt LLMs to generate intermediate reasoning steps. However, as the number of steps increases, errors tend to accumulate. (Madaan et al., 2023; Guan et al., 2024) introduce self-correction mechanisms that leverage feedback to refine incorrect reasoning and ensure accuracy. Moreover, Retrieval-Augmented Generation (RAG) (Xu et al., 2024; Wang et al., 2025b) and knowledge graphs (Wang et al., 2025a; Zhu et al., 2025) methods enhance reasoning by integrating structured external knowledge to improve accuracy and reduce hallucinations. To further enhance the performance through learning from interaction, recent work has shifted to supervised fine-tuning (SFT) (Zhang & Zhang, 2023) and reinforcement learning (RL) (Qian et al., 2025). In our work, we further introduce a task graph-based RL framework that enhances the model's long-horizon planning capability through reward signals derived from the task graph.

## 3 METHODOLOGY

As illustrated in Figure 1, GraphPlan mainly consists of three components: (a) High-level planning with task graph; (b) Memory-aware low-level action policy; (c) Dynamic replanning through scene graph memory. We provide the details of each component in this section.

### 3.1 HIGH-LEVEL PLANNING WITH TASK GRAPH

To provide physical grounding for high-level planning, we specifically design an automatic task graph construction pipeline to constrain the space of actions. The task graph serves three roles: it guides LLMs' planning via the prompt with the task graph, supports the design of reward functions in GRPO, and acts as an external verification mechanism to provide feedback for reasoning.

#### 3.1.1 TASK GRAPH CONSTRUCTION

We represent the constraint space of subtasks as a directed graph $\mathcal{G} = (\mathcal{V}, \mathcal{E})$, where nodes $\mathcal{V} = \{v_i\}_{i=1}^N$ denote subtasks of various types, and each edge $(v_i, v_j) \in \mathcal{E}$ encodes a permissible subtask transition, *i.e.* $v_j$ may follow $v_i$.

**Subtasks as Nodes.** Node $v_i$ representing subtask is denoted as $A_i(o_i)$ or $A_i(o_i, o_j)$, where $A_i$ denotes a high-level action and $o_i, o_j$ refer to relevant objects. To encourage the high-level planner to focus on object roles in subtasks, we use meta class $\mathcal{C} := \{\texttt{obj}, \texttt{rec}, \texttt{mov}, \texttt{spec}\}$ instead of the specific object name $o_i$, where $\texttt{obj}$ denotes non-container items, $\texttt{rec}$ designates fixed containers, $\texttt{mov}$ indicates portable containers, $\texttt{spec}$ encompasses items necessary for particular subtasks, e.g., 'fridge' for 'Cool'. During plan generation, meta classes are mapped to concrete object names from the environmental object set $\mathcal{O}_{\text{env}}$, which comprises all visible objects recognized by a pretrained object detection module. For convenience, we denote by $\mathcal{O}_{\text{env}}^c$ the set of environmental objects corresponding to each meta class $c$. Each subtask is mapped to a serialized low-level action policy that specifies the primitive actions required for task completion and is dynamically instantiated through environmental perception. Details can be found in the Appendix B.1. Following Min et al. (2021);

Yang (2024), we adopt 12 subtask nodes, which have been demonstrated to be an appropriate granularity to ensure that high-level planning can be reliably mapped to executable low-level actions. If necessary, subtask nodes can be expanded as the low-level policy evolves.

**Subtask Transition as Edges.** A task graph can be constructed by traversing training trajectories. However, we aim to establish it as a universal representation that generalizes beyond benchmark tasks. To achieve this, we automatically construct the graph by analyzing the functional compatibility between low-level action policies corresponding to subtasks. Specifically, we add directed edges $v_p \rightarrow v_q$ between nodes when the terminal low-level action of a preceding subtask $v_p$ (e.g., pickup) is functionally compatible with the initiating action of a subsequent subtask $v_q$ (e.g., put) within the agent's capability range. This mechanism enables both generalization to complex task combinations beyond training coverage and application to novel tasks without expert demonstrations.

**Planning objective.** Given a natural language instruction $\tau$, high-level planning aims to generate a subtask sequence $S(\tau) = (s_1, s_2, \ldots, s_T)$ that guides the agent to complete the task as instructed. Here, the subtask sequence should form a feasible path $\Pi = (v_1, v_2, \ldots, v_T)$ in $\mathcal{G}$, and each object in $s_i$ should belong to the environmental object set $\mathcal{O}_{\text{env}}$.

### 3.1.2 Prompt with Task Graph

To mitigate planning hallucinations in LLM-based task planning, we propose a novel prompting strategy incorporating robotic domain knowledge via a task graph. This strategy directs the agent to generate high-level subtask sequences strictly following the graph's topology, selecting each subsequent subtask only from the neighbors of the previous node. Specifically, the task graph nodes are encoded as action-target pairs assigned unique labels (*e.g.*, '[A] PickupObject(object)'), while the edges are converted into symbolic connection relationships (*e.g.*, 'A → B'). Additionally, we incorporate static prompt components, such as agent roles, planning rules, and the expected output format, to facilitate in-context learning. The complete prompt template is shown in Appendix E.

### 3.1.3 GRPO with Task Graph

Recent work (Qian et al., 2025; Vojnovic & Yun, 2025) demonstrates that rule-based rewards combined with RL significantly enhance reasoning. Since task planning allows multiple valid solutions, reward design is challenging without grounding. We address this with a task graph-based reward covering diverse feasible paths, integrated into GRPO optimization, along with format and instruction-following rewards to ensure plan reliability.

**Format Reward.** Prior research (Guo et al., 2025; Xie et al., 2025) has shown that the format reward $R_{\text{fmt}}$ effectively constrains the model's output to match the expected structure. The detailed definition of $R_{\text{fmt}}$ is provided in Appendix B.2.

**Node-level Reward.** To encourage each generated subtask $s_i$ is valid, we compute a node-level reward via dynamically weighted object-validity scores across four meta-classes in $\mathcal{C}$:

$$R_{\text{node}} = \sum_{c \in \mathcal{C}} w_c \cdot R_{\text{node}}^c, \tag{1}$$

where the weight $w_c$ is dynamically adjusted based on whether the corresponding meta class $c$ appears in the plan. Weights of unused meta classes are set to zero, and the remaining weights are renormalized to ensure that all weights sum to 1. For each meta class, the reward $R_{\text{node}}^c$ is defined as the valid rate of objects belonging to meta class $c$ in the plan, where an object is considered valid when its name exists in the corresponding environmental list $\mathcal{O}_{\text{env}}^c$, then:

$$R_{\text{node}}^c = \frac{1}{|\mathcal{O}_S|} \sum_{o \in \mathcal{O}_S} \mathbb{I}\left(o \in \mathcal{O}_{\text{env}}^c\right), \tag{2}$$

where $\mathcal{O}_S$ denotes the set of all objects in plan $S(\tau)$, and $\mathbb{I}(\cdot)$ is an indicator function that returns 1 if the condition is met and 0 otherwise.

**Edge-level Reward.** To encourage subtask transitions conforming to the edges $\mathcal{E}$ in the task graph $\mathcal{G}$, this reward traverses and checks all adjacent subtask pairs $(s_t, s_{t+1})$ in the generated plan. The reward is defined as follows:

$$R_{\text{edge}} = \prod_{1 \le t \le T-1} \mathbb{I}(s_t, s_{t+1} \in \mathcal{V}) \cdot \mathbb{I}((s_t, s_{t+1}) \in \mathcal{E}), \tag{3}$$

where the reward equals 1 if both subtasks $(s_t, s_{t+1})$ are nodes in $\mathcal{V}$ and the transition between them satisfies the edge connection of $\mathcal{G}$, and 0 otherwise.

**Instruction Following Reward.** The reward for instruction following $R_{\text{inst}}$ encourages the model to generate plans that satisfy user instructions. Unlike tasks with unique solutions (*e.g.*, arithmetic or classification), task planning is inherently multi-solution, making exact-match evaluation insufficient. To address this, we employ a two-stage reward based on the ground-truth (GT) plan $S^*$. First, we extract critical subtasks $\mathcal{S}_{\text{key}}^*$ from $S^*$ that are essential for completing the instruction. The generated plan $S(\tau)$ receives a reward of 0.5 if it covers all critical subtasks, and a full reward of 1.0 only if it exactly matches the GT plan $S^*$. Otherwise, the reward is 0. This two-stage design ensures that the model is first driven to cover indispensable steps and then incentivized to produce fully precise plans, even when multiple valid paths exist.

**Overall Reward Function.** The overall reward function integrates multiple reward components to guide the model in generating executable subtask sequences that meet instruction goals.

$$R_{\text{total}} = R_{\text{fmt}} + R_{\text{node}} + R_{\text{edge}} + R_{\text{inst}}. \tag{4}$$

### 3.1.4 VERIFICATION WITH TASK GRAPH

Inspired by Gou et al. (2023), we treat the task graph as an external validation tool and leverage its feedback to refine the responses of the LLM. The verification function is composed of two parts:

**Node-level Legitimacy Verification**. To determine whether each object in the initial subtask sequence is valid, we perform a node-level legitimacy verification: $\mathbb{V}_{\text{node}} = \prod_{c \in \mathcal{C}, o \in \mathcal{O}_S} \mathbb{I}(o \in \mathcal{O}_{\text{env}}^c)$.

**Edge-level Legitimacy Verification.** To verify that all action transitions satisfy the edge constraints of the task graph, the edge-level legitimacy check is defined analogously to the edge-level reward $R_{\text{edge}}$ (see Eq. (3)): $\mathbb{V}_{\text{edge}} = R_{\text{edge}}$.

For any invalid subtask or transition, the corresponding error message is fed back to guide LLM correction. This *Verify → Feedback → Correct* cycle iterates until the plan passes validation or reaches the maximum iteration limit (*i.e.* 3).

### 3.2 MEMORY-AWARE LOW-LEVEL ACTION POLICY

The low-level action policy is responsible for mapping high-level abstract subtasks into sequences of executable primitive actions, including both navigation and interaction components. These actions are dynamically instantiated through environmental perception. The agent first performs map-based path planning to navigate near the target object and faces it, then invokes Mask R-CNN (He et al., 2017) to obtain the instance segmentation mask of the object, and finally passes it to the interaction function for execution.

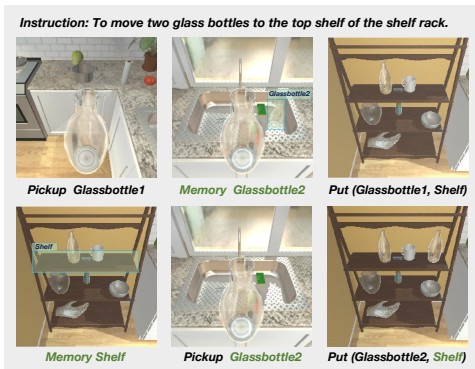

Figure 2: Qualitative case to show benefits of the low-level action policy with memory.

Following Yang (2024), a grid-based differentiable dynamic map is constructed from an egocentric RGB perspective, integrating navigable properties and affordance of objects (*e.g.*, 'openable') to support navigation. To enhance navigation efficiency and prevent redundant interactions, we develop a *memory-aware low level action policy*. While prior works (*e.g.*, (Kim et al., 2023)) have explored memory-enhanced low-level policy, our approach advances in proactive caching and instance-level discrimination.

Unlike methods that primarily record past interactions, our agent proactively detects and caches information for objects that may require future interaction during execution. When the agent encounters other task-related objects on the way to the primary target object, it logs its current location as a candidate waypoint and caches the corresponding segmentation mask, in addition to saving the position and mask of the target object. Thus, subsequent subtasks can directly utilize cached positions instead of re-exploring, improving navigation consistency and efficiency. Following a "best-view" policy, memory entries are updated only when a new observation offers a closer or more canonical viewpoint of the object. For distinguishing multiple instances of the same type, we maintain a

"latest interaction" record for each instance to track their dynamic state. This provides independent identifiers for similar-looking objects, enabling precise discrimination and consistent placement.

As shown in Figure 2, for tasks like moving two glass bottles to a shelf, this memory mechanism allows detecting and recording secondary 'Glassbottle2' during the execution of 'Pickup(Glassbottle1)'. Subsequently, when executing 'Put(Glassbottle2, Shelf)', the agent prioritizes the shelf position retrieved from memory instead of re-exploring the environment, ensuring the second bottle is placed on the same shelf level and substantially improving navigation efficiency and placement accuracy.

### 3.3 DYNAMIC REPLANNING

Although high-level planning with the task graph yields high-quality initial plans, it is not well grounded in dynamic environments and cannot readily adapt to newly emerged failures. To address this, we design an event-driven dynamic replanning module that is triggered by either low-level errors or high-level subtask completion, using a dynamic scene graph to provide environment context.

#### 3.3.1 SCENE GRAPH MEMORY

To support closed-loop reasoning and replanning, we maintain a dynamic scene graph $\mathcal{G}_{\text{scene}}$ as a task-aware environmental memory. Unlike a complete scene representation (Gu et al., 2024; Takmaz et al., 2025), $\mathcal{G}_{\text{scene}}$ is a compact, dynamically updated set of semantic triples $(o_i, r_{ij}, o_j)$ that describe relations $r_{ij} \in \mathcal{R}$ (*e.g.*, "on", "next to") between task-relevant objects, with key attributes (*e.g.*, color and shape) as node attributes. This design naturally limits graph scale while directing planning attention to environmentally critical cues, enabling more efficient and accurate reasoning. The $\mathcal{G}_{\text{scene}}$ construction process is as follows: (1) **Key Object Extraction.** A language model parses the instruction $\tau$ to obtain $\mathcal{O}_{\text{key}} \subseteq \mathcal{O}_{\text{env}}$. (2) **Viewpoint Capture.** During execution, record the agent's egocentric views that contain any object in $\mathcal{O}_{\text{key}}$. (3) **Triple Generation.** The captured RGB and segmentation maps are fed into a Vision Language Model (VLM) to produce relation triples and object attributes, where incorporating segmentation helps align labels and reduce ambiguity. (4) **Scene Graph Update.** New triples are incrementally merged into $\mathcal{G}_{\text{scene}}$ via a dynamic update strategy as exploration proceeds, allowing adaptation to object state changes and post-interaction environmental updates. The details are provided in Appendix B.3.

#### 3.3.2 EVENT-DRIVEN REPLANNING

We hope the agent can interleave execution with thinking, continuously evaluating plan feasibility against environmental feedback and making necessary adjustments. To this end, we design a LLM-based replanning module driven by dual events (*i.e.*, low-level action errors and high-level subtask completion), to address both execution failures and instruction misalignment (Figure 3). Within this module, the task graph is also leveraged to mitigate planning hallucinations in LLMs. At each activation, the model assesses the need to adjust the plan based on the original plan, current subtask, feedback, and scene graph. Execution continues if no adjustment is needed, or switches to the revised plan otherwise. See Appendix B.4 for details.

**Low-level Action Error.** This trigger handles concrete execution failures, detected via feedback signals from the environment (*e.g.*, interaction errors) or from the navigation policy (*e.g.*, exhaustive search failure). When such an error occurs, the reasoning LLM integrates the current scene observations with commonsense knowledge to revise the plan. For the instance in Figure 3, if a potato cannot be found for slicing, the LLM might suggest searching inside closed containers like a refrigerator or microwave. The detailed execution of this case is provided in Appendix D.1 Figure 8.

**High-level Subtask Completion.** This trigger detects and tackles plans that are executable but misaligned with instruction goals. The LLM assesses the current plan against the continuously updated scene graph memory and the plan execution progress to verify alignment with the final objective. For example, for the goal "place the lettuce on the microwave oven table," if the scene graph indicates the microwave is on a side table but the plan targets a countertop, the LLM identifies this mismatch and triggers a correction. This allows proactive rectification of high-level planning errors undetectable by low-level error feedback, using subtask completion as checkpoints.

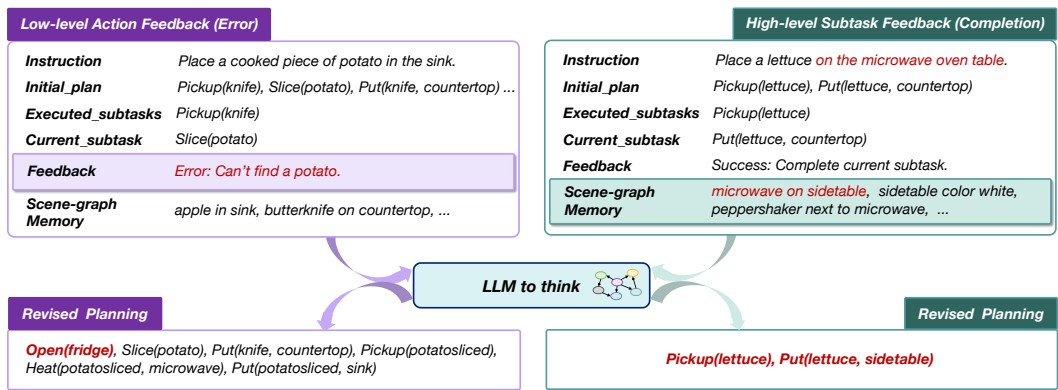

Figure 3: The two examples illustrate two types of event-driven dynamic replanning.

Table 1: Comparison with SOTA methods on ALFRED test set. We get the results of the baselines from the official ALFRED leaderboard or papers. 'Step-by-step instructions' and 'LLMs' columns, respectively, indicate whether step-by-step instructions and LLMs are used in high-level planning.

| Method | Step-by-step Instructions | LLMs | Tests Seen | | Tests Unseen | |
|---|---|---|---|---|---|---|
| | | | SR ↑ | GC ↑ | SR ↑ | GC ↑ |
| LLM-planner (Song et al., 2023) | ✓ | ✓ | 18.20 | 26.77 | 16.42 | 23.37 |
| CAPEAM (Kim et al., 2023) | ✓ | ✗ | 52.58 | 60.98 | 50.36 | 61.40 |
| DISCO (Yang, 2024) | ✓ | ✗ | 59.59 | 66.06 | 56.55 | 66.87 |
| Flare (Kim et al., 2025) | ✓ | ✓ | 40.05 | 48.84 | 40.88 | 51.72 |
| DISCO (Yang, 2024) | ✗ | ✗ | 58.05 | 64.96 | 54.77 | 65.56 |
| OPEx (Shi et al., 2024) | ✗ | ✓ | 43.51 | 54.27 | 41.27 | 53.82 |
| EPO (Zhao et al., 2024) | ✗ | ✓ | 64.79 | 72.30 | 62.35 | 67.52 |
| RoboGPT (Chen et al., 2025) | ✗ | ✓ | 59.92 | 67.83 | 62.00 | 72.09 |
| **GraphPlan** (ours) | ✗ | ✓ | **67.71** | **73.96** | **68.52** | **75.76** |

## 4 EXPERIMENTS

We first validate GraphPlan's hierarchical framework on ALFRED for vision-language navigation and interaction, then evaluate its high-level planner on ALFRED and unseen long-horizon tasks.

### 4.1 COMPARISON WITH SOTA ON ALFRED

**Benchmark and Metrics.** We conduct experiments on ALFRED (Shridhar et al., 2020), a challenging benchmark for robotics instruction following. The language instruction $L = (L_{\text{high}}, L_{\text{low}})$ contain both high-level goals and step-by-step guidance. ALFRED is partitioned into 'train', 'validation' and 'test' sets. Both 'validation' and 'test' are further divided into seen and unseen splits, where the unseen partitions consist of scenes absent from the training set. See Appendix C.1 for details. We adopt two evaluation metrics. The primary metric is Task Success Rate (SR), which measures the percentage of fully completed tasks. Additionally, Goal-Condition Success Rate (GC) evaluates the percentage of satisfied goal conditions. Test set results require leaderboard submission.

**Performance.** To demonstrate the effectiveness of our approach, we compare GraphPlan to competitive works in the test set reported on ALFRED public leaderboard. Following Kim et al. (2025); Chen et al. (2025), we report baselines using 1) only the goal instruction, and 2) both the goal instruction and step-by-step instructions. As shown in Table 1, our approach significantly outperforms prior works by 6.17% on unseen tasks, while achieving SOTA performance across all metrics in both unseen and seen environments (reaching 67.71% and 68.52%, respectively), demonstrating the effectiveness of our approach. Moreover, GraphPlan surpasses all previous methods in both using step-by-step instructions and even only the goal instruction, highlighting the superiority of our high-level planning module. It is worth emphasizing that GraphPlan does not rely on metadata for environmental grounding, indicating its stronger generalization capability to real-world scenarios.

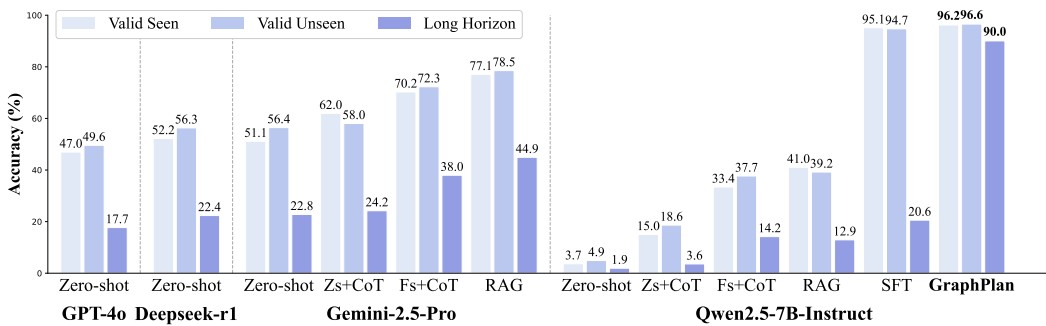

Figure 4: Success rates for high-level planning on ALFRED validation set and long-horizon tasks. For brevity, we adopt the abbreviations Zs for Zero-shot and Fs for Few-shot.

## 4.2 ABLATION STUDY ON THE HIERARCHICAL FRAMEWORK OF GRAPHPLAN

We conduct an ablation study of GraphPlan's components, with experimental results on the ALFRED validation set shown in Table 2. (1) Replacing our task graph-based planner (row 6) with a RAG-enhanced Qwen2.5-7B-Instruct planner (row 1) leads to significant performance drops of 27.66% and 27.11% on seen and unseen splits, respectively, underscoring the importance of high-quality initial planning. (2) Removing the replanning module results in open-loop execution that cannot adapt to unexpected errors or environmen-

Table 2: Ablation studies on GraphPlan.

| Component | Valid Seen | | Valid Unseen | |
|---|---|---|---|---|
| | SR ↑ | GC ↑ | SR ↑ | GC ↑ |
| planner replacer | 40.70 | 46.19 | 41.59 | 47.82 |
| w/o replan | 60.18 | 68.39 | 59.82 | 68.10 |
| w/o replan_low | 64.31 | 72.13 | 64.10 | 71.58 |
| w/o replan_high | 63.58 | 71.65 | 64.25 | 72.10 |
| w/o memory | 67.26 | 74.39 | 66.59 | 74.13 |
| GraphPlan | **68.36** | **75.32** | **68.72** | **76.01** |

tal feedback, reducing success rates by 8.18% and 8.90%. We then separately ablate the two types of event-driven replanning. Without low-level execution error driver (row 3), the agent fails to handle execution errors (e.g., wandering due to objects in closed containers or detection failures), reducing SR by 4.05% and 4.62%. Without high-level subtask completion driver (row 4), the agent cannot verify and correct plans against environmental information, causing SR drops of 4.78% and 4.47%. (3) Disabling the object state and location memory in the low-level policy (row 5) degrades performance by impairing object localization and introducing interaction errors. In tasks requiring placement of two identical objects into the same container, the agent may move one object repeatedly or misplace them in different locations, resulting in task failures. These results demonstrate that GraphPlan ensures robust execution through: task graph-based planning for high-quality initial plans, dynamic replanning for maintaining executability and instruction alignment, and memory-aware navigation for reliable interaction.

## 4.3 COMPARISON OF HIGH-LEVEL PLANNING

To evaluate our graph-enhanced task planning mechanism, we compare GraphPlan against multiple strong baselines, including API-based and open-source models under diverse inference settings. Since ALFRED contains limited and relatively simple task types, we construct a new long-horizon dataset containing 1396 samples, to better evaluate the ability of graph-enhanced planning on unseen long-horizon tasks. This dataset extends the original 7 short task types with 17 more complex long task types (details in Appendix C.2). For fair comparison, we consider three representative closed-source models (*i.e.*, GPT-4o, Deepseek-r1, and Gemini-2.5-Pro) to reflect the state of the art in general reasoning and language understanding. Besides, to enhance the adaptability of general-purpose agents to planning tasks, we employ common inference strategies (*i.e.*, zero-shot, CoT, few-shot, RAG) with the agents. As GraphPlan uses Qwen2.5-Instruct-7B, we also compare against its variants under different inference methods and SFT to assess our specific contributions.

**Metrics.** For high-level planning, success rate is the primary evaluation metric. Since multiple feasible paths often exist in task planning and certain GT annotations in datasets may not accurately match the instructions, relying solely on the single GT in the validation set is overly crude. Naturally, we incorporate the task graph as a verification tool and propose a more comprehensive validation method. The procedure begins with graph-constrained verification and GT matching. If both checks pass, the plan is successful. If graph verification fails, the plan is marked as failed. When the plan

satisfies graph constraints but diverges from GT, it undergoes further validation using an LLM.

**How does GraphPlan compare to state-of-the-art LLMs?** As shown in Figure 4, GraphPlan achieves superior performance across all settings, with 96.22% on valid seen, 96.56% on valid unseen and 90.04% on long horizon tasks. Although prompt-driven variants such as few-shot and RAG yield substantial improvements, GraphPlan still surpasses all such methods by a significant margin. **Does task graph enhancement improve long-horizon reasoning capability?** GraphPlan significantly outperforms the best prompt-based variant (*i.e.*, RAG), with a 56.27% average gain on validation sets and long-horizon success improving from 12.93% to 90.04%. While SFT achieves strong validation performance, its accuracy drops sharply to 20.57% on long-horizon tasks. This aligns with expectations: due to high similarity between ALFRED's training and validation sets, SFT handles in-domain short-term planning adequately but struggles with complex long-horizon tasks where planning hallucinations and missing steps prevail. In contrast, GraphPlan achieves 90.04% on long-horizon tasks, demonstrating that our task graph-based RL generalizes effectively without costly fine-tuning. For specific cases, see the Appendix D.2.

## 4.4 ABLATION ON HIGH-LEVEL PLANNER IN GRAPHPLAN

We conduct ablation studies on the three modules involving the task graph in the high-level planner to evaluate their individual contributions (Table 3): (1) **Prompt with Task Graph.** Removing the task graph from the prompt during inference significantly reduces planning accuracy, showing that the graph enhances the LLM's ability to generate grounded plans and reduces hallucinations. (2) **Graph GRPO with Task Graph.** Replacing the GRPO-trained model with the base Qwen-2.5-Instruct-7B results in

Table 3: Success rate contributions of individual components in the proposed planner.

| Method | Valid Seen | Valid Unseen | Long Horizon |
|---|---|---|---|
| w/o Graph p. | 77.93 | 74.11 | 40.69 |
| w/o GRPO | 49.88 | 52.64 | 33.67 |
| w/o Verification | 95.12 | 95.09 | 86.25 |
| All | 96.22 | 96.56 | 90.04 |

significantly degraded performance, which underscores the crucial role of our task graph-based reinforcement learning framework. (3) **Verification with Task Graph.** Removing verification feedback has minimal impact on the validation set, as initial responses mostly follow graph constraints. However, on long-horizon tasks where constraints are sometimes violated, error feedback improves performance by 3.79%. Even without refinement, our model outperforms others.

## 4.5 ABLATION ON KEY COMPONENTS OF GRPO

We conduct a quantitative ablation study to analyze key components in GRPO, as shown in Table 4. The "all" setting (row 4) indicates success rates with full rewards and the task graph prompt. Results here differ from Figure 4 as we removed validation feedback to isolate training effects. Further ablation studies are in Appendix C.8. Our findings are:

Table 4: Ablations of GRPO.

| Method | Comp. | Valid Seen | Valid Unseen | Long Horizon |
|---|---|---|---|---|
| w/o node+edge | reward | 93.05 | 89.33 | 44.63 |
| w/o instruction | reward | 84.27 | 84.66 | 55.59 |
| w/o graph p. | prompt | 85.00 | 83.44 | 59.46 |
| all | – | **95.12** | **95.09** | **86.25** |

**(1) Task graph reward enhances generalization and zero-shot learning:** Without the graph reward (Row 1), reliance on only the instruction reward restricts learning to task paths in the dataset. By contrast, the graph reward encourages exploration of all feasible subtask transitions, as the task graph encapsulates diverse possible paths. Its absence hinders long-horizon planning, confirming that the graph reward significantly improves generalization to complex task combinations beyond training coverage. Even without the instruction reward (Row 2), using only unlabeled data and graph rewards achieves an average 84.47% on the valid set, demonstrating the graph reward's ability to guide high-level action learning and impart basic planning capabilities in zero-shot scenarios. **(2) Instruction reward aligns user intent:** Without the instruction following reward, the model may engage in reward hacking by solely generating plans that satisfy graph constraints while disregarding actual instruction requirements. These results underscore the essential role of the instruction following reward in ensuring semantic alignment between the generated plans and the original task instruction. **(3) Incorporating the task graph into prompts enhances training effectiveness:** When we remove the explicit task graph from the prompt while keeping all other training and inference settings unchanged (Row 3, "w/o graph p."), performance drops markedly on both the validation set and long-horizon tasks (average declines of 10.89% and 26.79%,

respectively). These results confirm that the task graph not only aids model reasoning but also substantially improves training effectiveness.

## 5 GENERALIZATION TO NOVEL TASKS

To further validate node scalability and generalization beyond ALFRED's subtasks, we have expanded the task graph with 9 additional skills supported by the underlying AI2-Thor simulator but not used in ALFRED: BreakObject, FillObject, PushObject, PullObject, ThrowObject, DropObject, DirtyObject, CookObject, and UseUpObject. Following a similar construction methodology as the Long-Horizon Dataset, we have created 10 new task types combining these novel skills (e.g., Break&Throw, Clean&Fill&Heat/Cool&Place), with 400 samples. For example, the "Dirty&Clean&Break&Cool&Place" task requires the agent to "dirty a bowl and then clean it. next, put an egg into the bowl, break it, and refrigerate the final product." Detailed execution steps in the scene are shown in Figure 5.

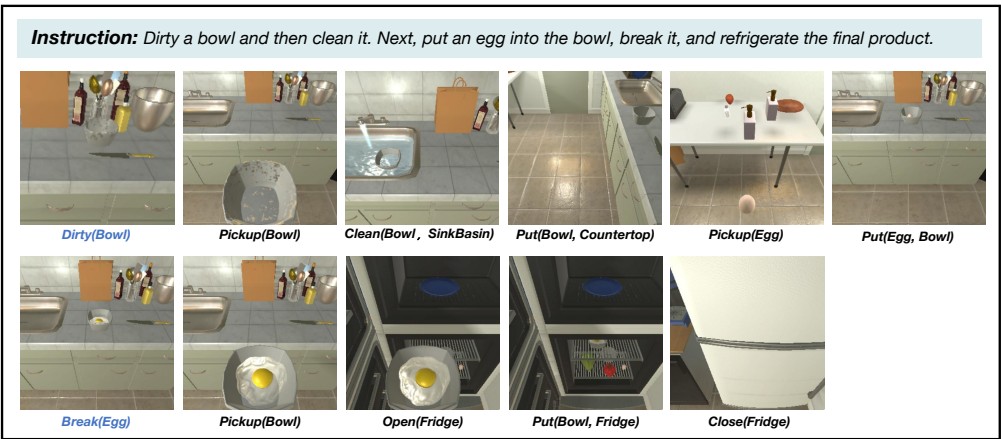

Figure 5: An Example of GraphPlan on newly added tasks. The new tasks 'DirtyObject' and 'BreakObject' are highlighted in blue.

By extending the task graph with new functional constraints, we evaluate GraphPlan's zero-shot generalization to these novel tasks. As shown below, GraphPlan achieves remarkable performance, while the baselines perform poorly:

Table 5: Generalization performance across new skills.

| Method | Model | Long Horizon | New Tasks |
|---|---|---|---|
| RAG | Gemini-2.5-Pro | 44.91 | 48.75 |
| SFT | Qwen-2.5-Instruct-7b | 20.57 | 16.25 |
| GraphPlan | Qwen-2.5-Instruct-7b | **85.96** | **80.50** |

## 6 CONCLUSION

We introduce GraphPlan, a general graph-enhanced planning framework that improves LLM-based embodied task planning. Our method reduces planning hallucinations by grounding reasoning in the structured task graph and ensuring plan executability through event-driven replanning with dynamic scene graphs. Experimental results on ALFRED validate the effectiveness of our components, while tests on a new long-horizon dataset demonstrate strong reasoning and generalization capabilities. Additional experiments show promising zero-shot learning potential. Future work could extend this approach to asynchronous planning or multi-robot collaborative systems by augmenting edges with temporal and state constraints, or by constructing multi-layer graphs.

ETHICS STATEMENT

This work presents a planning framework for embodied AI agents, evaluated entirely in simulated environments using the public ALFRED benchmark. We affirm compliance with the ICLR Code of Ethics. Our research does not involve human subjects, personal data, or real-world interactions, thus mitigating concerns regarding privacy, security, and safety. The work is based on standard synthetic datasets and does not focus on or exacerbate social biases. No conflicts of interest are declared.

REPRODUCIBILITY STATEMENT

To ensure the reproducibility of our work, we provide a detailed description of the GraphPlan framework's key components and implementation details, including task construction, scene graph updating, and the design of the reinforcement learning reward function. In Appendix C.3, we present a comprehensive implementation description, covering full training details (e.g., base models, hyperparameters for RL and SFT, and computational resources), the architecture of the high-level planner and replanning module, and specifications of the navigation components. The source code will be made publicly available upon acceptance.

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

# A   THE USE OF LLMs

The use of Large Language Models (LLMs) in this work was limited to linguistic refinement to enhance the clarity and readability of this paper. All scientific ideas and content remain entirely the responsibility of the authors. All text has been reviewed and approved by the authors.

# B   METHOD DETAILS

## B.1   DETAILS OF TASK GRAPH

In the task graph, each node $v_i$ corresponds to a subtask and is represented as $A_i(o_i)$ or $A_i(o_i, o_j)$, where $A_i$ is a high-level action and $o_i, o_j$ are associated objects. We use four meta-classes rather than specific object names to guide the high-level planner toward emphasizing object roles. During plan generation, each meta-class is instantiated by selecting concrete objects from the environmental object set according to the instruction requirements. The specific nodes are shown in Figure 6.

Table 6: The definition of nodes in the task graph.

| Node | High-level Action | Target |
|------|-------------------|--------|
| A | PickupObject | object |
| B | ToggleObject | lamp |
| C | SliceObject | object |
| D | PutObject | (object, receptacle) |
| E | FindSecond | object |
| F | PickSecond | object |
| G | PutPickObject | (object, movable_receptacle) |
| H | CoolObject | (object, Fridge) |
| I | HeatObject | (object, Microwave) |
| J | CleanObject | (object, SinkBasin) |
| K | OpenObject | receptacle |
| L | CloseObject | receptacle |

High-level sub-tasks are mapped to a serialized low-level action policy that specifies the primitive actions required for task completion and is dynamically instantiated by through environmental perception. For example, the subtask Heat(object, microwave) maps to a low-level policy (i.e., Navigate(microwave), OpenObject(microwave), PutObject(object, microwave), CloseObject(microwave), ToggleObjectOn(microwave), ToggleObjectOff(microwave), OpenObject(microwave), PickupObject(object), CloseObject(microwave)). Here, 'Navigate' requires path planning according to the environment, while the execution of 'Open' depends on the current open state of the microwave. The low-level actions comprise two categories: navigation actions (`RotateRight`, `RotateLeft`, `MoveAhead`, `LookUp`, `LookDown`) and interaction actions (`PickupObject`, `PutObject`, `OpenObject`, `CloseObject`, `ToggleObjectOn`, `ToggleObjectOff`, `SliceObject`). Importantly, some sub-tasks may appear identical to low-level actions, but the difference is that when (Pickup, apple) is a sub-task, the agent typically must perform multiple navigation actions until it is sufficiently close to the apple before the interactive action 'Pickup' can be executed.

## B.2   DETAILS OF GRPO REWARDS

**Format Reward.** We customized the format reward to suit the output structure of high-level task planning. The format reward is computed as:

$$R_{\text{fmt}} = \begin{cases} 1.0, & \text{if XML and JSON valid,} \\ 0.5, & \text{if XML valid but JSON invalid,} \\ 0.0, & \text{otherwise.} \end{cases} \tag{5}$$

$R_{\mathrm{fmt}} = 0.5$, if the model's response follows `XML_format`, *i.e.*, *<think>* ... *</think><answer>* ... *</answer>*; $R_{\mathrm{fmt}} = 1$, only when the content inside *<answer>* ... *</answer>* is a `JSON_object` including the specified key, *e.g.*, "high_level".

### B.3 DETAILS OF SCENE GRAPH

The scene graph is designed to provide task-specific environmental cues for the event-driven replanning module. It is generated dynamically from the agent's egocentric views via a VLM during execution, without requiring preconstruction, and supports real-time updates.

Compared to prior works, our scene graph construction method differs and improves in terms of task relevance and dynamic construction:

(i) General scene graphs (Gu et al., 2024; Takmaz et al., 2025) contain numerous task-irrelevant objects and relations, which can overwhelm LLMs with excessive environmental details during planning. Our scene graph is specifically designed as a task-focused memory module, not a complete environment map. By storing only task-relevant objects, its size remains manageable (typically under 20 nodes in ALFRED, despite scenes containing around 68 object instances). This design naturally limits graph scale and complexity while directing planning attention to environmentally critical cues, enabling more efficient and accurate reasoning.

(ii) Scene graphs are often generated statically (Takmaz et al., 2025; Wu et al., 2021) and cannot adapt to object state changes or post-interaction environment updates. Some planning methods rely on pre-built scene graphs (Rana et al., 2023; Liu et al., 2025a), yet such ideally constructed graphs are often unavailable in real-world settings. Therefore, we adopt a dynamic construction pipeline based on the VLM and further ensure reliability through complementary mechanisms:

To enhance VLM-based scene capture quality, we process both RGB and segmentation inputs to ensure label alignment and reduce ambiguity, while also implementing preventive measures through carefully designed few-shot prompts and rule-based constraints (e.g., requiring physical plausibility for spatial relationships). Our dynamic update mechanism maintains scene graph accuracy through pruning strategies, pose updates, and instance differentiation during environmental interactions. Even with incomplete captures from certain viewpoints, multi-view observations accumulate over time, thus ensuring comprehensive coverage of task-relevant objects.

### B.4 DETAILS OF EVENT-DRIVEN REPLANNING

Our event-driven replanning framework is designed to address general categories of failures in embodied tasks: low-level execution errors and high-level semantic misalignment. This design is not specific to the ALFRED benchmark but applies to any domain where an agent must recover from actionable failures or correct logically valid but incorrect plans.

**Failure Detection Mechanisms.** The detection of a "failure" differs for the two trigger events, combining rule-based signals with semantic reasoning:

(i) Rule-based detection for low-level errors: the agent relies on explicit feedback signals. For instance, the environment reports interaction failures (e.g., "slice action with fork failed"), and our navigation module throws predefined exceptions when target objects cannot be located after an exhaustive search.

(ii) LLM-based reasoning for semantic alignment: to detect "successful execution but wrong goal" failures, we utilize the reasoning power of LLMs. Upon completing a subtask, the LLM evaluates the current state against the scene graph memory. For example, if the instruction is "place lettuce on the microwave table" but the plan targets a countertop, the LLM infers the mismatch by checking the scene graph (which locates the microwave on a side table) and triggers a correction. This allows proactive rectification of high-level planning errors that low-level signals cannot catch.

**Key Distinctions from Prior Works.** The main differences between our event-driven replanning and prior works are as follows:

(i) Our approach combines reactive error handling with proactive checkpoint assessment at subtask completion to address low-level execution errors and intent alignment, while Huang et al. (2022);

Kim et al. (2025); Chen et al. (2025) rely on low-level execution failures and PRED (Kim et al., 2024b) uses predefined state differences.

(ii) We employ dual-graph constraints where the task graph mitigates planning hallucinations and the scene graph focuses environmental reasoning, significantly reducing hallucinations, while prior methods (Huang et al., 2022; Kim et al., 2024b; 2025; Chen et al., 2025) depend on LLM free-generation.

(iii) Our dynamic scene graph structures environmental memory with task-relevant object relationships and attributes, providing richer semantic context than raw perception history (Huang et al., 2022; Chen et al., 2025) or object-level difference lists (Kim et al., 2024b).

These design choices ensure that the replanning module is robust, generalizable, and capable of correcting a broad spectrum of planning failures in embodied task execution.

## C  EXPERIMENT DETAILS

### C.1  DETAILS OF ALFRED

ALFRED is divided into train, validation, and test sets, with the validation and test sets further split into seen and unseen subsets depending on whether the corresponding scenes appear in the training phase. The dataset includes 7 task types, 58 target object classes, and 26 receptacle classes distributed across 120 indoor scenes. Objects within the same class often exhibit various visual appearances (*e.g.*, there are 30 varieties of apples), and the indoor scenes cover kitchens, bathrooms, bedrooms, and living rooms. According to the official statistics, the training set consists of 21023 examples, the validation seen set contains 820 examples, the validation unseen set contains 821 examples, the test seen set has 1533 examples, and the test unseen set has 1529 examples.

### C.2  CONSTRUCTION OF LONG HORIZON DATASET

While ALFRED contains limited and relatively simple task types (7 base types + 5 sliced variations): Pick & Place, Stack & Place, Pick Two & Place, Clean & Place, Heat & Place, Cool & Place, Examine in Light, we construct a new long-horizon dataset with 1396 samples to better evaluate graph-enhanced planning capabilities on unseen complex tasks. This dataset extends the original 7 task types into 17 more complex long-horizon categories.

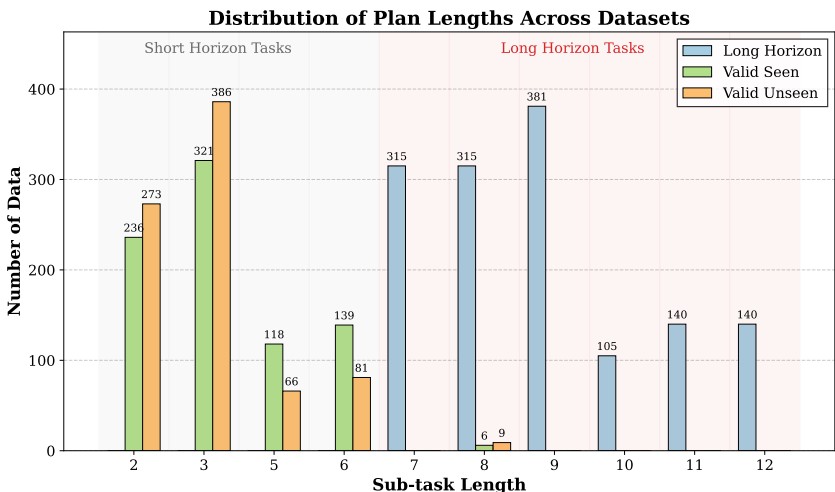

Figure 6: Comparison of plan lengths across ALFRED validation sets and long horizon data.

The new task types include multi-step compositions such as: Pick Two & Stack & Heat & Place; Pick Two & Stack & Cool & Place; Pick Two & Clean & Stack & Place; Stack & Cool & Place; Stack & Heat & Place; Stack & Clean & Place; Slice & Heat & Stack & Place; Slice & Cool &

Stack & Place; Slice & Clean & Stack & Place; Slice & Clean & Place & Heat & Place; Slice & Clean & Place & Cool & Place; Slice & Clean & Place & Clean & Place; Pick Three & Cool & Place; Pick Three & Heat & Place; Pick Three & Clean & Place; Slice & Clean & Stack & Place & Heat & Stack & Place; Pick Two & Place & Slice & Cool & Stack & Place.

Under the current subtask configuration, ALFRED's maximum subtask length is 8, while our long-horizon dataset increases the average subtask length from 3.4 to 8.9, with a maximum of 12 subtasks. The comparison between the ALFRED validation set and our long-horizon dataset is shown in Figure 6.

**Construction of Long Horizon Dataset.** The dataset is constructed through: (1) generating new task types via basic task combination; (2) designing high-quality seed instructions for each new task type using meta-classes instead of specific objects; (3) employing LLMs to paraphrase and abstractly describe seed instructions for diversity; (4) referencing previous templated planning methods to design high-level action templates with fixed action sequences and variable object parameters; (5) generating a new long-horizon dataset with ground-truth plans by instantiating (instruction, action sequence) pairs using ALFRED environment objects to populate meta-classes and action template parameters. Furthermore, **we conducted manual verification to ensure the quality and correctness of the generated instructions and their corresponding action sequences.**

## C.3 Implementation details

**Training Details.** For RL experiments, we use Qwen2.5-7B-Instruct as the base model and TRL (von Werra et al., 2020) as our training framework. The training configuration includes: 2 epochs, batch size of 48, learning rate of $1 \times 10^{-6}$, KL coefficient of 0 (Liu et al., 2025b), and rollout number of 12. All experiments are conducted on 6 NVIDIA A100 GPUs, with consistent hyperparameters across RL and ablation studies. For supervised fine-tuning, we train Qwen-2.5-7B-Instruct on the training set for 3 epochs using AdamW optimizer with a peak learning rate of $1 \times 10^{-6}$ and linear learning rate scheduling.

**High-level Planning.** The high-level subtask planning module uses Qwen2.5-7B-Instruct model after RL training, while GPT-4o serves as the reasoning model for replanning.

**Navigation Details.** Use the navigator from Yang (2024), with Mask R-CNN (He et al., 2017) serving as the core model for object detection and instance segmentation. The model provides: (1) object class labels and bounding boxes in RGB frames, using the training-time object list as the environmental object list; (2) binary instance masks for target localization and object interaction during navigation. Depth estimation and functional properties are handled by two separate U-Nets, with functional properties including one navigation category and seven interaction categories.

## C.4 Comparison with Graph Search Methods

Table 7: Performance comparison with graph search methods

| Strategy | LLM | Valid Seen | Valid Unseen | Long Horizon |
|---|---|---|---|---|
| Greedy Search | Qwen-2.5-Instruct-7B | 19.02 | 19.39 | 2.58 |
| Greedy Search | DeepSeek-R1-Distill-Qwen-32B | 46.34 | 49.08 | 12.82 |
| Beam Search | Qwen-2.5-Instruct-7B | 17.93 | 18.16 | 1.86 |
| Beam Search | DeepSeek-R1-Distill-Qwen-32B | 40.70 | 43.68 | 11.39 |
| GraphPlan | Qwen-2.5-Instruct-7B | **96.22** | **96.56** | **90.04** |

A natural alternative to our LLM-based planning approach would be to employ direct graph search algorithms on the predefined task graph. However, we identify fundamental limitations in applying graph search to high-level task planning. First, nodes in our task graph represent abstract task types defined by high-level actions and object meta-classes, not concrete subtasks. This abstraction requires semantic grounding to map meta-classes to instruction-specific objects before execution by downstream policies, which is a capability lacking in pure graph search. Furthermore, while graph search can explore possible paths, it requires a model to score candidate nodes at each step. This necessitates numerous LLM calls, incurring high computational costs and creating strong dependence on the scoring model's local accuracy.

To empirically validate these limitations, we conduct experiments using depth-first search with greedy/beam strategies and LLM-based node scoring, as shown in Table 7. Both greedy and beam search strategies achieve significantly lower success rates across all evaluation sets compared to GraphPlan, regardless of whether we use Qwen-2.5-Instruct-7B or the larger DeepSeek-R1-Distill-Qwen-32B for node scoring.

We attribute this performance gap to two key factors. First, graph search makes locally optimal decisions at each step, leading to myopic error accumulation (Yao et al., 2023) and suboptimal global paths. Second, the planning quality is fundamentally limited by scoring accuracy at each node. This also explains why DeepSeek-R1-Distill-Qwen-32B significantly outperforms Qwen-2.5-Instruct-7B in graph search settings, as the larger model provides more reliable scoring. This also explains why DeepSeek-R1-Distill-Qwen-32B significantly outperforms Qwen-2.5-Instruct-7B in graph search settings, as the larger model provides more reliable scoring.

## C.5 TASK GRAPH CONSTRUCTION AND COMPLEXITY ANALYSIS

### C.5.1 TASK GRAPH CONSTRUCTION ANALYSIS

To address concerns regarding the sensitivity of task graph quality to training data and its applicability to tasks without expert demonstrations, we conduct a systematic analysis of different graph construction strategies, including full-data traversal, few-shot sampling, pure functional compatibility, and hybrid methods. The results are shown in Table 8:

Table 8: Planning success rates under different task graph construction methods.

| Construction Method | Valid Seen | Valid Unseen | Long Horizon |
|---|---|---|---|
| Full Dataset Traversal (No Expansion) | 95.98 | 94.23 | 52.22 |
| Few-shot Sampling (98 samples) | 95.49 | 94.48 | 52.01 |
| Functional Compatibility Only | **96.59** | 95.83 | 89.76 |
| Few-shot + Functional Expansion | 96.22 | **96.56** | **90.04** |

Our results demonstrate that: (1) the task graph quality is relatively robust to training data quantity, as long as the available data covers all basic task types. (2) The task graph can be constructed solely through functional compatibility analysis without expert demonstrations. (3) Using functional compatibility for graph construction or completion is crucial for generalizing to complex long-horizon tasks, as graphs relying only on training trajectories perform poorly on "Long Horizon" (52.22% vs. 90.04%).

### C.5.2 TASK GRAPH COMPLEXITY ANALYSIS

To analyze the impact of task graph complexity, we expanded the original 12-node graph by incorporating 9 additional nodes representing AI2-THOR capabilities unused in ALFRED, creating extended graphs with 16 and 21 nodes respectively. We then evaluated GraphPlan's performance across task graph of different sizes, with success rates summarized below:

Table 9: Performance comparison with different task graph sizes.

| # Nodes | Valid Seen | Valid Unseen | Long Horizon |
|---|---|---|---|
| 12 | 95.12 | 96.93 | 90.83 |
| 16 | 94.88 | 95.71 | 86.68 |
| 21 | 93.29 | 94.60 | 85.96 |

The experimental results show only minor performance degradation even with increased graph complexity, demonstrating GraphPlan's strong scalability and robustness.

## C.6 PERFORMANCE ANALYSIS ACROSS VARYING TASK HORIZONS

To comprehensively evaluate the robustness and scalability of our approach, we analyze GraphPlan's performance across tasks of increasing complexity (7–12 subtasks), comparing it against two strong baselines: Gemini-2.5-Pro enhanced with RAG and a SFT model based on the same Qwen2.5-7B-Instruct backbone. The results are shown in Table 10, Table 11, Table 12 respectively. In the table, all metrics represent average values, with the following abbreviations: Acc (Accuracy), GPR (Graph Pass Rate), MSR (Missing Step Rate), ASR (Additional Step Rate), WTR (Wrong Transfer Rate), and AER (Affordance Error Rate).

Table 10: Performance of GraphPlan with increasing task horizons.

| Length | Acc (%) | GPR (%) | MSR (%) | ASR (%) | WTR (%) | AER (%) | Inference Time (s) | LLM Calls |
|---|---|---|---|---|---|---|---|---|
| 7 | 98.73 | 99.68 | 0.96 | 1.14 | 0.02 | 0.12 | 7.68 | 1.05 |
| 8 | 96.51 | 98.41 | 1.12 | 0.65 | 0.06 | 0.00 | 7.82 | 1.10 |
| 9 | 85.04 | 96.33 | 1.98 | 1.44 | 0.45 | 0.00 | 9.05 | 1.21 |
| 10 | 86.67 | 96.19 | 0.54 | 5.71 | 0.26 | 0.00 | 9.34 | 1.2 |
| 11 | 81.43 | 95.71 | 1.90 | 8.81 | 0.38 | 0.00 | 9.64 | 1.21 |
| 12 | 80.71 | 85.00 | 5.90 | 4.19 | 2.16 | 0.10 | 23.70 | 2.10 |

Table 11: Performance of Gemini-2.5-Pro with RAG with increasing task horizons.

| Length | Acc (%) | GPR (%) | MSR (%) | ASR (%) | WTR (%) | AER (%) | Inference Time (s) | LLM Calls |
|---|---|---|---|---|---|---|---|---|
| 7 | 72.06 | 83.17 | 13.51 | 10.81 | 1.56 | 0.15 | 226.88 | 1.00 |
| 8 | 70.16 | 83.81 | 12.38 | 10.20 | 1.90 | 0.00 | 210.78 | 1.00 |
| 9 | 29.40 | 62.47 | 19.58 | 10.83 | 5.40 | 0.40 | 214.25 | 1.00 |
| 10 | 30.48 | 53.33 | 15.00 | 23.75 | 4.51 | 1.25 | 221.46 | 1.00 |
| 11 | 15.71 | 80.00 | 38.00 | 22.45 | 2.44 | 0.00 | 293.69 | 1.00 |
| 12 | 9.29 | 65.00 | 38.33 | 66.67 | 0.91 | 0.42 | 335.96 | 1.00 |

Table 12: Performance of SFT with increasing task horizons.

| Length | Acc (%) | GPR (%) | MSR (%) | ASR (%) | WTR (%) | AER (%) | Inference Time (s) | LLM Calls |
|---|---|---|---|---|---|---|---|---|
| 7 | 40.63 | 82.86 | 29.92 | 37.45 | 2.21 | 0.32 | 7.65 | 1.00 |
| 8 | 41.90 | 81.27 | 30.32 | 29.65 | 2.89 | 0.10 | 7.36 | 1.00 |
| 9 | 4.20 | 16.01 | 26.76 | 3.96 | 11.20 | 1.62 | 7.16 | 1.00 |
| 10 | 5.71 | 20.00 | 30.00 | 39.82 | 6.42 | 0.14 | 8.99 | 1.00 |
| 11 | 0.00 | 19.29 | 28.10 | 58.81 | 6.36 | 1.28 | 9.38 | 1.00 |
| 12 | 3.57 | 6.43 | 19.24 | 0.00 | 3.98 | 6.18 | 9.05 | 1.00 |

(i) **Performance Scaling**: GraphPlan shows minimal degradation with longer horizons, with accuracy declining from 98.73% to 80.71% while graph pass rates remain above 85%. This contrasts sharply with baselines: Gemini-2.5-Pro with RAG drops from 72.06% to 9.29%, and SFT collapses from 40.63% to near zero beyond 9 steps.

(ii) **Reasoning Fidelity**: Following [1], we employ four fine-grained error metrics to identify specific weaknesses in LLM planning. Quantitative analysis using fine-grained error metrics reveals GraphPlan's superior planning quality. It achieves remarkably low wrong transfer rates (0.02-2.16%) and near-zero affordance errors, demonstrating accurate action-object understanding through graph constraints. While missing and additional step rates show some increase (e.g., additional steps: 1.14% to 8.81%), they remain substantially lower than baselines (e.g., Gemini's 10.81% to 66.67%).

(iii) **Computational Costs**: GraphPlan maintains stable inference times ( 7-9s) for 7-11 steps, demonstrating well-controlled overhead. Even at 12 steps, time only increases to 23.7s, significantly better than Gemini-2.5-Pro with RAG (210-336s). Despite verification mechanisms, LLM calls average only approximately 1.2, indicating most plans pass initial validation. These results collectively validate GraphPlan's robustness, efficiency, and reasoning quality across increasingly complex tasks.

Table 13: Execution Efficiency Analysis on the test set.

| Method | Tests Seen | | Tests Unseen | |
|--------|------------|---|--------------|---|
| | PLWSR ↑ | PLWGC ↑ | PLWSR ↑ | PLWGC ↑ |
| *Step-by-step Instructions* | | | | |
| CAPEAM (Kim et al., 2023) | 23.0 | 27.10 | 21.59 | 25.31 |
| DISCO (Yang, 2024) | 40.67 | 47.47 | 36.51 | 44.52 |
| Flare (Kim et al., 2025) | 16.68 | 21.31 | 18.14 | 22.78 |
| *w/o Step-by-step Instructions* | | | | |
| DISCO (Yang, 2024) | 39.66 | 46.59 | 35.50 | 43.64 |
| OPEx (Shi et al., 2024) | 13.64 | 20.13 | 12.57 | 18.46 |
| EPO (Zhao et al., 2024) | 56.92 | 66.20 | 51.99 | 64.15 |
| RoboGPT (Chen et al., 2025) | 29.97 | 35.44 | 33.61 | 38.66 |
| **GraphPlan** (ours) | 42.91 | 46.14 | 45.47 | 48.66 |

## C.7 EXECUTION EFFICIENCY ANALYSIS

While our primary evaluation focuses on planning success (SR and GC) to validate our hierarchical framework's effectiveness, execution efficiency offers a complementary perspective critical for practical deployment. To assess this quantitatively, we employ Path-Length Weighted Success Rate (PLWSR) and Path-Length Weighted Goal-Conditioned Success Rate (PLWGC). The results in Table 13 demonstrate that our method achieves competitive efficiency compared to several baselines.

First, compared to methods employing replanning mechanisms (Chen et al., 2025; Kim et al., 2025), GraphPlan achieves significantly higher PLWSR and PLWGC, indicating that our dual-graph constrained replanning produces more efficient corrections with minimal redundant steps. Second, when compared against DISCO (Yang, 2024), which shares the same underlying navigation policy, GraphPlan maintains comparable efficiency metrics despite incorporating additional replanning modules. This confirms that our substantial improvement in success rates is not achieved at the expense of excessive execution steps.

We note that the maximum step limit of 1000 in our experimental setup affects absolute PLW values, as failed episodes continue executing until reaching this limit. Future work may focus on integrating advanced goal-oriented navigation algorithms to further improve path efficiency while maintaining the robust planning capabilities demonstrated here.

## C.8 FURTHER ABLATION OF GRPO

**Reward Components.** To evaluate each reward in GRPO, we further conduct an ablation study on the node-level reward, edge-level reward and format reward, as shown in Table 14.

**Node and Edge Rewards are Essential:** Removing either the node reward-level (Row 1) or the edge-level reward (Row 2) results in performance decline. Notably, the absence of the edge reward causes a more significant drop, indicating that constraining subtask transitions through graph edges is crucial for grounded subtask decomposition. (3) **Format Reward Supports Structural Output Consistency:** When the format reward is ablated (Row 3), a slight performance drop is observed. This suggests that although other rewards can partially guide the model to produce structured outputs, explicit format supervision helps ensure structure correctness.

Table 14: Success rate of reward design and prompting strategies during the training stage.

| Method | Valid Seen | Valid Unseen | Long Horizon |
|--------|------------|--------------|--------------|
| w/o node | 93.54 | 91.66 | 76.50 |
| w/o edge | 89.88 | 85.03 | 50.50 |
| w/o format | 91.71 | 89.82 | 70.63 |
| all | **95.12** | **95.09** | **86.25** |

**Training Data.** To evaluate the impact of training data volume, we train the planning model using subsets corresponding to 1%, 10%, 20%, and 50% of the full training samples. These subsets cover all seven task types to ensure a fair representation of the training set. In Figure 7, the results indicate that model performance improves with increasing amounts of training data. The performance gain is relatively modest on the validation set, whereas a more significant improvement is observed on long-horizon tasks as the training data increases. This indicates that our planning method effectively strengthens short-task (*i.e.*, the validation) planning even with limited data, while scaling the training data helps LLMs learn richer action space required for long-horizon tasks.

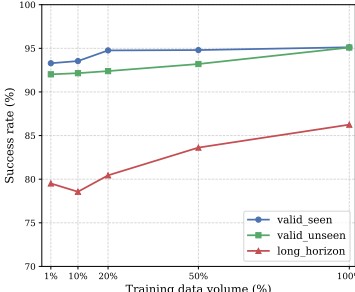

Figure 7: Ablation studies on data volume.

# D  CASE STUDY

## D.1  QUALITATIVE RESULTS OF REPLANNING

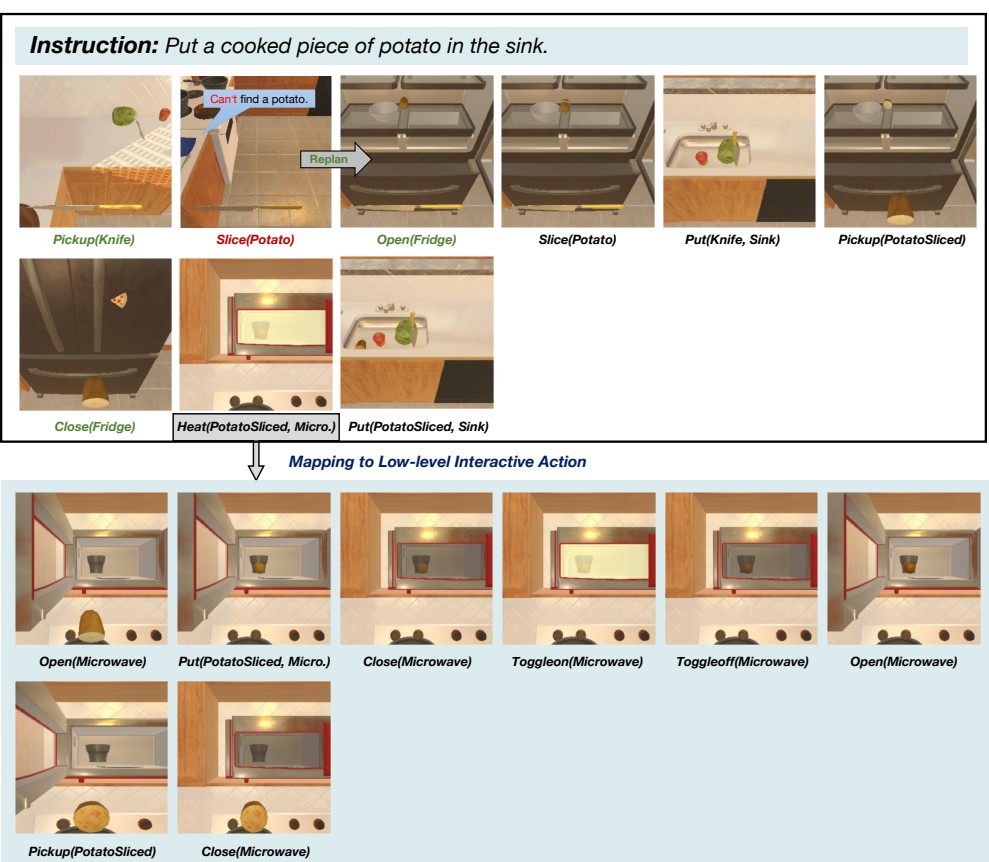

Figure 8: An Example of GraphPlan on replanning: When the agent fails to locate a potato during environmental exploration while executing slice(potato), an error triggers the replanning process. The revised plan then attempts to open the refrigerator to search for one.

## D.2 CASE STUDY ON LONG-HORIZON TASKS

Table 15: Comparison of long-horizon task planning performance with different training methods. "GRPO with GT" means to conduct GRPO training solely by using the instructions in the training set, and the ground truth plan.

| Case 1 |
|---|
| **Instruction:** Put a cleaned tomato slice in a bowl on the top shelf, and put a pan containing a heated potato on the dining table. |
| **GraphPlan (Ours):** (PickupObject, Knife), (SliceObject, Tomato), (PutObject, (Knife, SinkBasin)), (PickupObject, TomatoSliced), (CleanObject, (TomatoSliced, SinkBasin)), (Put-PickObject, (TomatoSliced, Bowl)), (PutObject, (Bowl, Shelf)), (PickupObject, Potato), (HeatObject, (Potato, Microwave)), (PutPickObject, (Potato, Pan)), (PutObject, (Pan, DiningTable))  
 **Assessment:** All instruction requirements are fully satisfied (e.g., *cleaned* tomato, *heated* potato), all objects are correctly placed *(tomato slice in bowl on shelf, potato in pan on table)*, and the action sequence follows valid transitions. |
| **GRPO with GT:** (PickupObject, Knife), (PickupObject, Potato), (HeatObject, (Potato, Microwave)), (PutObject, (Potato, Pan)), (PickupObject, Knife), (PickupObject, Tomato), (SliceObject, Tomato), (PutObject, (Knife, SinkBasin)), (PickupObject, TomatoSliced), (CleanObject, (TomatoSliced, SinkBasin)), (PutObject, (TomatoSliced, Bowl)), (PutObject, (Bowl, Shelf)), (PutObject, (Pan, DiningTable))  
 **Assessment:** Invalid transition: *(PickupObject, Knife) → (PickupObject, Potato), (PickupObject, Knife) → (PickupObject, Tomato), (PutObject, (TomatoSliced, Bowl)) → (PutObject, (Bowl, Shelf) → (PutObject, (Pan, DiningTable))*; Unable to perform consecutive pick or put. |
| **SFT:** (PickupObject, Knife), (SliceObject, Tomato), (PutObject, (TomatoSliced, Bowl)), (PutObject, (Bowl, Shelf)), (PickupObject, Potato), (HeatObject, (Potato, Microwave)), (Put-PickObject, (Potato, Pan)), (PutObject, (Pan, DiningTable))  
 **Assessment:** Invalid transition: *(PutObject, (TomatoSliced, Bowl)) → (PutObject, (Bowl, Shelf))*; Missing steps: lack putting the knife and cleaning the tomato slice. |
| Case 2 |
| **Instruction:** Put two potato slices in the pan, pick pan to fridge to cool and then put them on the counter. |
| **GraphPlan (Ours):** (PickupObject, Knife), (SliceObject, Potato), (PutObject, (Knife, SinkBasin)), (PickupObject, PotatoSliced), (FindSecond, PotatoSliced), (PutObject, (PotatoSliced, Pan)), (PickSecond, PotatoSliced), (PutObject, (PotatoSliced, Pan)), (Pick-upObject, Pan), (CoolObject, (Pan, Fridge)), (PutObject, (Pan, CounterTop))  
 **Assessment:** All instruction requirements are fully satisfied, correctly handle multiple objects *(two potato slices)*, and the action sequence follows valid transitions. |
| **GRPO with GT:** (PickupObject, Knife), (SliceObject, Potato), (PutObject, (Potato, SinkBasin)), (PickupObject, PotatoSliced), (FindSecond, PotatoSliced), (PutObject, (PotatoSliced, Pan)), (PickupObject, Pan), (PutPickObject, (Pan, Fridge)), (PutObject, (Pan, CounterTop))  
 **Assessment:** Incorrect steps: the agent should place the knife held in hand into the sink but erroneously placed the potato instead, and use *PutPickObject* incorrectly for cooling. |
| **SFT:** (PickupObject, Knife), (SliceObject, Potato), (PutObject, (Knife, countertop)), (Pick-upObject, PotatoSliced), (PutPickObject, (PotatoSliced, Pan)), (PutObject, (Pan, Microwave)), (CoolObject, (Pan, Fridge)), (PutObject, (Pan, CounterTop))  
 **Assessment:** Invalid transition: *(PutObject, (Pan, Microwave)) → (CoolObject, (Pan, Fridge))*. Missing steps: only put one potato slice in the pan. |

Reasoning results for case 1 with different training methods.

**Instruction:**

```
Put a cleaned tomato slice in a bowl on the top shelf, and put a
    pan containing a heated potato on the dining table.
```

--------------------------------------------------------------------

**GraphPlan:**

```
<reasoning>
To complete the task, we need to follow these steps:

1. **Pick up the knife** to slice the tomato.
2. **Slice the tomato** into a slice.
3. **Clean the tomato slice** in the sink basin.
4. **Put the cleaned tomato slice in a bowl** and pick up the bowl
    **.
5. **Place the bowl with the the cleaned tomato slice on the shelf
    **.
6. **Pick up the potato**.
7. **Heat the potato** in the microwave.
8. **Put the heated potato in a pan**.
9. **Place the pan with the heated potato on the dining table**.

We will validate each step against the action graph and ensure all
     actions are valid and follow the constraints.
</reasoning>

<answer>
{
    "high_level": [
        ["PickupObject", "Knife"],
        ["SliceObject", "Tomato"],
        ["PutObject", ["Knife", "SinkBasin"]],
        ["PickupObject", "TomatoSliced"],
        ["CleanObject", ["TomatoSliced", "SinkBasin"]],
        ["PutPickObject", ["TomatoSliced", "Bowl"]],
        ["PutObject", ["Bowl", "Shelf"]],
        ["PickupObject", "Potato"],
        ["HeatObject", ["Potato", "Microwave"]],
        ["PutPickObject", ["Potato", "Pan"]],
        ["PutObject", ["Pan", "DiningTable"]]
    ]
}
</answer>
```

--------------------------------------------------------------------

**GRPO with GT:**

```
<reasoning>
The task involves two main steps:
1. Cleaning a tomato, slicing it, and placing it in a bowl on the
    shelf.
2. Heating a potato, putting it in a pan, and placing the pan on
    the dining table.

Let's break down the steps for each part:

### Step 1: Clean Tomato, Slice, and Place in Bowl
1. **PickupObject(Knife)**: To use the knife for slicing.
2. **SliceObject(Tomato)**: To cut the tomato into slices.
```

```
3. **PutObject(Knife, SinkBasin)**: Put the knife back in the sink.
4. **PickupObject(TomatoSliced)**: Pick up the sliced tomato.
5. **CleanObject(SinkBasin)**: Clean the tomato slice in the sink.
6. **PutObject(TomatoSliced, Bowl)**: Place the cleaned tomato
   slice in the bowl.
7. **PutObject(Bowl, Shelf)**: Place the bowl on the dining table.

### Step 2: Heat Potato, Put in Pan, and Place on Dining Table
1. **PickupObject(Knife)**: To use the knife for slicing (if needed
   , but not necessary here).
2. **PickupObject(Potato)**: Pick up the potato.
3. **HeatObject([Potato, Microwave])**: Heat the potato in the
   microwave.
4. **PutObject(Potato, Pan)**: Place the heated potato in the pan.
5. **PutObject(Pan, DiningTable)**: Place the pan on the dining
   table.

We need to ensure that the actions follow the constraints and the
    high-level action graph.
</reasoning>
<answer>
{
    "high_level": [
        ["PickupObject", "Knife"],
        ["PickupObject", "Potato"],
        ["HeatObject", ["Potato", "Microwave"]],
        ["PutObject", ["Potato", "Pan"]],
        ["PickupObject", "Knife"],
        ["PickupObject", "Tomato"],
        ["SliceObject", "Tomato"],
        ["PutObject", ["Knife", "SinkBasin"]],
        ["PickupObject", "TomatoSliced"],
        ["CleanObject", ["TomatoSliced", "SinkBasin"]],
        ["PutObject", ["TomatoSliced", "Bowl"]],
        ["PutObject", ["Bowl", "Shelf"]],
        ["PutObject", ["Pan", "DiningTable"]]
    ]
}
</answer>
```

**SFT:**

```
"high_level": [
    ["PickupObject", "Knife"],
    ["SliceObject", "Tomato"],
    ["PutObject", ["TomatoSliced", "Bowl"]],
    ["PutObject", ["Bowl", "Shelf"]],
    ["PickupObject", "Potato"],
    ["HeatObject", ["Potato", "Microwave"]],
    ["PutPickObject", ["Potato", "Pan"]],
    ["PutObject", ["Pan", "DiningTable"]]
]
```

# E    PROMPTS

---

Prompt with Task Graph

```
# Role: Create a high-level plan based on the provided instruction,
    which uses the actions and objects allowed in the environment
    to complete the Alfred household task.

## Profile
- OBJECTS = [AlarmClock, Apple, AppleSliced, ArmChair, BaseballBat,
     BasketBall, BathtubBasin, Bed, Book, Bowl, Box, Bread,
     BreadSliced, ButterKnife, CD, Cabinet, Candle, Cart, CellPhone,
     Cloth, CoffeeMachine, CoffeeTable, CounterTop, CreditCard, Cup,
     Desk, DeskLamp, DiningTable, DishSponge, Drawer, Dresser, Egg,
     Faucet, FloorLamp, Fork, Fridge, GarbageCan, Glassbottle,
     HandTowel, Kettle, KeyChain, Knife, Ladle, Laptop, Lettuce,
     LettuceSliced, Microwave, Mug, Newspaper, Ottoman, Pan, Pen,
     Pencil, PepperShaker, Pillow, Plate, Plunger, Pot, Potato,
     PotatoSliced, RemoteControl, Safe, SaltShaker, Shelf, SideTable,
     SinkBasin, SoapBar, SoapBottle, Sofa, Spatula, Spoon,
     SprayBottle, Statue, StoveBurner, TennisRacket, TissueBox,
     Toilet, ToiletPaper, ToiletPaperHanger, Tomato, TomatoSliced,
     Vase, Watch, WateringCan, WineBottle, Lamp]
- RECEPTACLES = [ArmChair, BathtubBasin, Bed, Cabinet, Cart,
     CoffeeMachine, CoffeeTable, CounterTop, Desk, DiningTable,
     Drawer, Dresser, Fridge, GarbageCan, Microwave, Ottoman, Safe,
     Shelf, SideTable, SinkBasin, Sofa, StoveBurner, Toilet,
     ToiletPaperHanger, Bowl, Box, Cup, Mug, Plate, Pan, Pot]
- MOVABLE_RECEPTACLES = [Bowl, Box, Cup, Mug, Plate, Pan, Pot]
- KNIFE = [Knife, ButterKnife]
- LAMP = [DeskLamp, FloorLamp, Lamp]
- HIGH_LEVEL_ACTIONS = [PickupObject, PutObject, PutPickObject,
     CleanObject, HeatObject, CoolObject, ToggleObject, SliceObject,
     FindSecond, PickSecond, OpenObject, CloseObject]

## High-level Action Causality Graph
### Node Definitions
- [A] Action: PickupObject, Target: object
- [B] Action: ToggleObject, Target: lamp
- [C] Action: SliceObject, Target: object
- [D] Action: PutObject, Target: (object, receptacle)
- [E] Action: FindSecond, Target: object
- [F] Action: PickSecond, Target: object
- [G] Action: PutPickObject, Target: (object, movable_receptacle)
- [H] Action: CoolObject, Target: (object, Fridge)
- [I] Action: HeatObject, Target: (object, Microwave)
- [J] Action: CleanObject, Target: (object, SinkBasin)
- [K] Action: OpenObject, Target: receptacle
- [L] Action: CloseObject, Target: receptacle

### Directed Connections
A --> B
A --> D
A --> E
A --> G
A --> H
A --> I
A --> J
A --> K
A --> L
B --> A
B --> B
B --> C
B --> D
B --> E
```

```
B --> F
B --> G
B --> H
B --> I
B --> J
B --> K
B --> L
C --> C
C --> D
C --> G
C --> H
C --> I
C --> J
C --> K
C --> L
D --> A
D --> E
D --> F
D --> K
D --> L
E --> D
E --> F
F --> D
F --> G
F --> H
F --> I
F --> J
G --> D
G --> H
G --> I
G --> J
G --> K
G --> L
H --> D
H --> G
H --> J
I --> D
I --> G
I --> J
J --> D
J --> G
J --> H
J --> I
K --> A
K --> C
K --> D
K --> G
K --> L
L --> A
L --> C
L --> D
L --> G
L --> K
### Structural Features
   Parallel Branches from A:
   - B (ToggleObject)
   - D (PutObject)
   - G (PutPickObject)
   - H (CoolObject)
   - I (HeatObject)
   - J (CleanObject)
```

```
## Planning Rules

### Graph Constraints
- Actions must follow the allowed transitions in the high-level
    action graph.
### Valid Targets
- All targets must be from 'OBJECTS', 'RECEPTACLES, or
    MOVABLE_RECEPTACLES' lists.
- Replace unlisted targets with semantically similar ones from the
    lists.
### Object Interaction Rules
- The picked object must be put down before PickupObject can be
    executed again.
- When PutObject, make sure the robot have already PickupObject
    before.
- For placing two identical objects into a container: 'PickupObject
    ', 'FindSecond'(to locate the distinct second instance), '
    PutObject'(to place the first object), 'PickSecond', 'PutObject
    '(again to place the second object).
### Slicing Rules
- When slicing something with knife, follow this action sequence:
    PickupObject(Knife) -> SliceObject(Object) -> PutObject(Knife,
    Receptacle).
- After slicing, put the knife down and switch to the object_sliced
    (e.g., 'Tomato' to 'TomatoSliced'). Pickup object_sliced and
    put it in a receptacle, or put it in a receptacle after some
    high-level action (e.g., CoolObject, HeatObject or CleanObject).
### Stack Rules
- PutPickObject(object, movable_receptacle) means putting the
    picked object into a movable receptacle and then pick them
    together to put on the final static receptacle. Action sequence:
     'PickupObject(object)' -> 'PutPickObject(object,
    movable_receptacle)' -> 'PutObject(object, receptacle)'.
    Replaceable action sequence:'PickupObject(object)' -> 'PutObject
    (object, movable_receptacle)' -> 'PickupObject(
    movable_receptacle)' -> 'PutObject(object, receptacle)'.
### Instruction Semantics
- Complex numbers like "vases" or "two objects" are treated as 2 by
    default.

## Output Requirements
    Respond in the following format:
    <reasoning>
    ...
    </reasoning>
    <answer>
    ...
    </answer>
    Within <reasoning> tags, two phase include analysing the task
        step-by-step and then conducting the planning Validation.
    Within <answer> tags, keep JSON strictly. Generate plans in
        this EXACT structure:
    {
        "high_level": [["PickupObject", "Tomato"], ["PutObject", ["
            Tomato", "SinkBasin"]]]
    }
```

