# OpenReview forum: "GraphPlan: Graph-enhanced Planning via Thinking LLMs for Embodied Agents"
_ICLR.cc/2026/Conference — Submitted to ICLR 2026_

### Official Review · Reviewer_ADQz · 2025-10-28

**Soundness:** 3
**Presentation:** 3
**Contribution:** 1
**Rating:** 2
**Confidence:** 4

**Summary:**

The paper proposes GraphPlan, a graph-enhanced framework for embodied agents that integrates structured knowledge into LLM-based planning. It builds a task graph to constrain and verify language-generated plans, a scene graph to represent updating object relations, and a memory-aware low-level policy with replanning capability. It combines these components with an event-driven replanning mechanism and a reinforcement learning objective that rewards graph-consistent actions. The method is validated in the ALFRED benchmark and outperforms previous state-of-the-art methods by notable margins.

**Strengths:**

- The paper tackles challenging and important issues for long-horizon planning.
- The proposed method outperforms previous state-of-the-art methods with large margins.
- Incorporating subtask structure in graph forms is straightforward sensible.

**Weaknesses:**

- For task graph construction, the authors use four types of meta classes, but their motivation is not well described. Why should be they? In addition, the authors use subtasks from a specific benchmark. Can they generalize to novel tasks?
- For task graph construction and its verification, the task graph is constructed from expert trajectories. How sensitive is the quality of the task graph to the number and diversity of the training samples? Quantitative analyses should be conducted. In addition, can the proposed method be used in case of tasks without training demos?
- For memory-aware low-level action policy, logging waypoints and object masks has already been explored (e.g., Kim et al, 2023). What are the core differences from the prior work? And, why are they significant?
- The proposed event-driven replanning is highly motivated by several failure modes in a specific benchmark, raising a concern of their generalizability.
- Replanning has been actively explored [1,2], but little to no discussion about them is provided. What are the main difference with them?
  - Huang et al., "Inner monologue: Embodied reasoning through planning with language models," CoRL, 2022.
  - Kim et al, "Pre-emptive Action Revision by Environmental Feedback for Embodied Instruction Following Agents," CoRL, 2024.
- It is unclear if the comparison with prior work in Table 1 is fair. For example, how many training samples are used compared to prior work? Some prior work (Song et al., 2023, Kim et al, 2025) uses only a small portion of the training samples (~100 samples). In addition, is the comparison done under the same LLMs?
- The evaluation is conducted in a single benchmark, raising a generalizability concern. Can this approach be applied to other task setups?

**Questions:**

- In Sec. 3.3.2, how does the agent know if it fails at a task? Is it predicted by LLMs or measured by some predefined rules?
- Due to the discrete nature of graphs, it might be hard to apply the proposed method to tasks with continuous state and action spaces. Can the proposed method be used for such tasks?

---

> ### Author Response · Authors · 2025-11-23
>
> > **W1 and W7**: Clarify the motivation for meta-classes in task graph construction and the method's generalizability to novel tasks.
>
> Thank you for these questions regarding meta-class design and generalization.&#x20;
>
> **Motivation for Meta-classes**&#x20;
>
> The use of meta-classes encourages the planner to **focus on functional properties and generalize across objects with similar affordances**. Using specific object names would lead to an impractically large graph, while this design allows the same graph structure to adapt to various environmental objects. For instance, in *PutPick(object, movable\_receptacle)*, the meta-classes represent portable containers and destinations without relying on concrete names.
>
> **Generalization to Novel Tasks**
>
> We have systematically evaluated generalization through two complementary approaches:
>
> First, as reported in **Sec. 4.3**, we tested generalization to complex task compositions by creating a new dataset with 17 long-horizon task types using existing ALFRED subtasks, where GraphPlan achieves strong performance.
>
> Second, to further validate node scalability and generalization beyond ALFRED's original subtasks, we expand the Task Graph with 9 additional skills supported by the underlying AI2-Thor simulator but not used in ALFRED: BreakObject, FillObject, PushObject, PullObject, ThrowObject, DropObject, DirtyObject, CookObject, and UseUpObject. Following a similar construction methodology as the Long-Horizon Dataset, we have created 10 new task types combining these novel skills (*e.g., Break\&Throw, Clean\&Fill\&Heat/Cool\&Place*), with 400 samples. For example, a Dirty\&Clean\&Break\&Cool\&Place task requires the agent to "Dirty a bowl and then clean it. Next, put an egg into the bowl, break it, and refrigerate the final product". Detailed execution steps of this example in the scene are shown in **Sec. 5** **Figure 5**.
>
> By extending the Task Graph with new functional constraints, we evaluate GraphPlan's zero-shot generalization to these novel tasks. As shown in Table S1 below, benefiting from the structured guidance of task graph, **GraphPlan remains effective on novel tasks**, whereas the baselines significantly underperform.
>
> Table S1: Success rates for high-level planning on long-horizon tasks and new tasks.
>
> | Method    | Model                | Long Horizon | **New Tasks** |
> | --------- | -------------------- | ------------ | ------------- |
> | RAG       | Gemini-2.5-Pro       | 44.91        | 48.75         |
> | SFT       | Qwen-2.5-Instruct-7b | 20.57        | 16.25         |
> | GraphPlan | Qwen-2.5-Instruct-7b | **85.96**    | **80.50**     |
>
> > **W2**: Sensitivity of task graph quality to training data and applicability to tasks without demonstrations.
>
> Thank you for these insightful questions. We would like to clarify that our task graph construction is **relatively insensitive** to training data quantity and can adapt to tasks without demonstrations through functional constraints. While initial graph construction can utilize template-based methods \[1,2] or training trajectories, our goal is to enable task graphs to serve as *universal representations within a given environment*. To achieve this, we automatically expand the graph by analyzing functional compatibility between subtasks. Specifically, we add directed edges between nodes when the terminal low-level action of a preceding subtask (*e.g., pickup*) is functionally compatible with the initiating action of a subsequent subtask (*e.g., put*) within the agent's capability range. **This functional expansion mechanism enables both generalization to complex task combinations beyond training coverage and application to novel tasks with no expert demonstrations.** Table S2 quantitatively validates the planning performance of GraphPlan across different task graph construction strategies:
>
> **Table S2:** Planning success rates (%) across different task graph construction methods
> | Construction Method | Valid Seen | Valid Unseen | Long Horizon |
> |---------------------|-------------|--------------|-------------|
> | Full Dataset Traversal (No Expansion)| 95.98 | 94.23 | 52.22 |
> | Few-shot Sampling (98 samples) | 95.49 | 94.48 | 52.01 |
> | Functional Compatibility Analysis Only   | 96.59       | 95.83        | 89.76        |
> | Few-shot Sampling + Functional Expansion | 96.22 | 96.56 | **90.04** |
>
> The results show: (1) task graph quality is robust to training data if basic task types are covered; (2) task graph can be built solely via functional compatibility analysis without expert demonstrations; (3) Using functional compatibility for graph construction or completion is key to generalizing to complex long‑horizon tasks (52.22% vs. 90.04%).
>
> \[1] Min et al. "Film: Following instructions in language with modular methods." *arXiv,&#x20;*&#x32;021.
>
> \[2] Xu et al. "DISCO: Embodied Navigation and Interaction via Differentiable Scene Semantics and Dual-level Control." *ECCV*, 2024.

---

> ### Author Response · Authors · 2025-11-23
>
> > **W3**: The core differences and significance of the memory mechanism compared to prior works (e.g., Kim et al, 2023).
>
> Thank you for this suggestion and we have added related discussion in our revised manuscript (**Sec. 3.2**). While prior works like CAPEAM \[3] have explored memory-enhanced low-level policy, our approach advances in two aspects: proactive caching and instance-level discrimination.
>
> Unlike methods that primarily record past interactions, our agent **proactively detects and caches information** for objects that may require future interaction during execution. When the agent encounters other task-related objects on the way to the primary target object, it logs its current location as a candidate waypoint and caches the corresponding segmentation mask, in addition to saving the position and mask of the target object. Thus, subsequent subtasks can directly utilize cached positions instead of re-exploring, improving navigation consistency and efficiency. Following a "best-view" policy, memory entries are updated only when a new observation offers a closer or more canonical viewpoint of the object.
>
> For distinguishing multiple instances of the same type, while existing methods like \[3] employ an exclusion strategy that removes processed objects from the semantic map, our method **maintains a "latest interaction" record** for each instance to track their dynamic state. This provides independent identifiers for similar-looking objects, enabling precise discrimination and consistent placement.
>
> \[3] Kim et al. "Context-aware planning and environment-aware memory for instruction following embodied agents." ICCV, 2023.
>
> > **W4 and Q1**:  The event-driven replanning seems tailored to specific failures in ALFRED. How generalizable is it? How does the agent detect task failure, via LLMs or predefined rules?
>
> Thank you for this question and we wish to clarify that our event-driven replanning framework is designed to address general categories of failures in embodied tasks: (i) **low-level execution errors** (*e.g.*, navigation and interaction failure), and (ii) **semantic misalignment where actions are executable but do not satisfy the instruction** (*e.g.*, placing an object on the wrong surface).&#x20;
>
> The notion of "failure" is different for the two types of triggers:&#x20;
>
> **(i) Rule-based detection for low-level errors**: The agent relies on explicit feedback signals. For instance, the environment reports interaction failures (*e.g.*, "slice action with fork failed"), and our navigation module throws predefined exceptions when target objects cannot be located after an exhaustive search.
>
> **(ii) LLM-based reasoning for semantic alignment**: To detect "successful execution but wrong goal" failures, we utilize the reasoning power of LLMs. Upon completing a subtask, the LLM evaluates the current state against the scene graph memory. For example, if the instruction is "place lettuce on the *microwave table*" but the plan targets a *countertop*, the LLM infers the mismatch by checking the scene graph (which locates the microwave on a side table) and triggers a correction. This allows proactive rectification of high-level planning errors that low-level signals cannot catch.

---

> ### Author Response · Authors · 2025-11-23
>
> > **W5**: Clarify the key differences between GraphPlan's replanning and prior works.
>
> Thanks for your recommendation. While our initial manuscript already discussesed some replanning work, following your suggestion, we have added discussion with \[4,5]. The main differences are as follows:
>
> (i) Our approach combines reactive error handling with **proactive checkpoint assessment at subtask completion** to address low-level execution errors and **intent alignment**, while Inner Monologue \[4] relies on low-level execution failures and PRED \[5] uses predefined state differences.
>
> (ii) We employ dual-graph constraints where **the task graph mitigates planning hallucinations** and the scene graph focuses environmental reasoning, significantly reducing hallucinations, while both methods \[4,5] depend on LLM free-generation.
>
> (iii) Our **dynamic scene graph&#x20;**&#x73;tructures environmental memory with **task-relevant&#x20;**&#x6F;bject relationships and attributes, providing richer semantic context than raw perception history \[4] or object-level difference lists \[5].
>
>
> \[4] Huang et al, "Inner monologue: Embodied reasoning through planning with language models," CoRL, 2022.
>
> \[5] Kim et al, "Pre-emptive Action Revision by Environmental Feedback for Embodied Instruction Following Agents," CoRL, 2024.
>
> > **W6**: Clarify the fairness of comparisons in Table 1 regarding training data size and model consistency. Some prior work (Song et al., 2023, Kim et al, 2025) uses only a small portion of the training samples (\~100 samples).
>
> Thank you for this question. Following established evaluation practices \[2,6,7], Table 1 primarily demonstrates our framework's SOTA effectiveness on the official leaderboard for complete instruction following tasks. We acknowledge the importance of fair comparisons and addressed this through multiple experimental dimensions:
>
> (i) To fairly evaluate our main contribution (*i.e.*, graph-enhanced high-level planning). We compared GraphPlan against leading API-based LLMs, as well as SFT and standard GRPO **with the same backbone**, demonstrating consistent advantages. Additionally, extensive ablation analyses validated the contribution of each module in our approach.&#x20;
>
> (ii) Our original analysis in **Appendix C.8** already demonstrated robustness using progressively smaller subsets (50% to 1% of training data). Following your suggestion, we have specifically evaluated performance with only 98 training samples, evenly sampled from all seven task types to ensure representative coverage. The comprehensive results are shown below:
>
> **Table S3:** Planning success rates (%) across different training data volumes.
>
> | Data Volume  | 98    | 210   | 2,100 | 4,200 | 10,500 | 21,023 |
> | ------------ | ----- | ----- | ----- | ----- | ------ | ------ |
> | Valid Seen   | 92.56 | 93.29 | 93.54 | 94.76 | 94.80  | 95.12  |
> | Valid Unseen | 91.78 | 92.02 | 92.15 | 92.39 | 93.20  | 95.09  |
>
> These results confirm our approach's **remarkable data efficiency and few-shot capability.** The minimal performance difference (≤2.5%) between using only 98 samples and the full dataset demonstrates that data volume has negligible impact on Table 1 comparisons, verifying that GraphPlan's advantages stem from task graph structural knowledge rather than training scale or model disparities.
>
> \[6] Zhao et al. "EPO: Hierarchical LLM Agents with Environment Preference Optimization." EMNLP, 2024.
>
> \[7] Chen et al. "Robogpt: an llm-based long-term decision-making embodied agent for instruction following tasks." TCDS, 2025.
>
> > **Q2**: Can the method handle continuous state and action spaces despite graph discretization due to the discrete task graph?
>
> Thank you for this question. Although the task graph is discrete, GraphPlan's core framework is hierarchical, enabling it to handle continuous state and action spaces through the low-level policy. This decoupling allows discrete planning and continuous execution. The task graph provides structured knowledge to guide the LLM in decomposing complex tasks into semantic subtask sequences (*e.g.*, "*PickupObject(object)*", "*OpenObject(receptacle)*"). The low-level policy is responsible for grounding these discrete subtasks into continuous actions (*e.g.*, motion planning, force control).&#x20;

---

> ### Author Response · Authors · 2025-11-27
>
> Dear Reviewer ADQz,
>
> As the discussion period comes to a close, we would greatly appreciate it if you could kindly take a moment to review our responses at your convenience. If there are any remaining points that would benefit from further clarification, we would be more than happy to address them before the discussion window closes. Thank you again for your thoughtful review and for the time you've dedicated to our work.
>
> Sincerely,
>
> The Authors

---

### Official Review · Reviewer_JYqU · 2025-11-01

**Soundness:** 3
**Presentation:** 3
**Contribution:** 2
**Rating:** 6
**Confidence:** 4

**Summary:**

The paper proposed GraphPlan, a framework that incorporates a Task Graph and a Scene Graph for embodied agent task completion in the ALFRED environment. The task graph is a detailed representation of task-specific related information, including objects, subtask relationships, attributes, affordance, etc. It aims to help high level planning, generate reward function for action executions, and provide task verification for reasoning. The method also includes a memory for task relevant object states and locations through waypoints and segmentations. The Scene graph is generated on the fly to store key objects (extracted by LLMs), view points of the objects, and the relationship among objects. It aims to help replanning when a low level action error or subtask planning error is encountered. The paper benchmarked the performance in comparison against multiple methods and demonstrated improvement on task success rate and goal condition success rate. The paper also conducted thorough ablation studies to showcase the importance of each components in the proposed framework. Further more, the paper collected a new datasets of 1396 samples to evaluate long-horizon high level planning capabilities in comparison against 3 closed-source models and varied prompting strategies. The proposed method demonstrated outstanding performance especially among long horizon tasks.

**Strengths:**

- The paper is well structured and well written.
- The paper proposed a method for embodied agent planning and action prediction, leveraging a proposed task graph and a scene graph
- The paper incorporated a replanning stage to improve long horizon task performance in case of errors
- The paper conducted thorough experiments among baselines, evaluating the proposed method through task success rate, goal condition success rate, and numerous ablation studies.
- The paper also collected a dataset specifically aiming at evaluating long horizon tasks, and conducted experiments against closed-source LLMs for high level planning

**Weaknesses:**

1. The paper is careful to highlight the that "the subtask nodes can be expended as the low-level policy evolves" and that "the proposed task graph-based approach can generalize to planning tasks in other domains". While they are indeed possible, the main question is really 'How useful is the method outside the ALFRED/simulated environment? In cases such as general home robot, takeout delivery robot, or disaster rescue robot, how feasible is it to manually generate the task-specific graphs, annotate all the key subtasks, and exhaust all possible/potential states/objects/attributes/conditions/relationships/subtasks?'
2. Section 3.2: it was a bit unclear how exactly low-level action policies are designed & trained when incorporating a memory of object states a locations?
3. While evaluating the performance, the main metrics are Task Success Rate (SR) and Goal-Conditioned Success Rate (GC). Is task execution efficiency (e.g. #actions/#steps it took to reach a goal) a relevant metrics to consider perhaps?

**Questions:**

a. It is great that the scene graphs can be generated automatically on the fly. How truthful are the generated scene graphs?
b. Figure 4: is there any reason why most of the models perform better in the valid unseen dataset compared to the valid seen dataset?

---

> ### Author Response · Authors · 2025-11-23
>
> > **W1**: How feasible and scalable is the task graph approach for real-world applications (*e.g.*, home/delivery/rescue robots) given the potential need for manual graph construction?
>
> We appreciate this practical concern about real-world deployment. While real-world integration requires careful system design, our hierarchical planning framework provides a more scalable and verifiable foundation than end-to-end alternatives.
>
> **First, the task graph can be automatically expanded through functional compatibility analysis.** We systematically add directed edges between nodes when a preceding subtask's terminal low-level action (*e.g., pickup*) is functionally compatible with a subsequent subtask's initiating action (*e.g., put*) within the agent's capability range. This enables generalization to novel task combinations without exhaustive manual annotation, as validated by our newly added experiments on novel skills in **Sec. 5**. For different robotic systems, practitioners can construct task graphs by: identifying the system's available primitive skills; selecting appropriate subtask granularity aligned with these skills; and establishing transitions based on functional constraints.&#x20;
>
> **Second, our hierarchical design effectively bridges discrete planning with continuous execution.** While the task graph operates at the semantic level (*e.g., "ReceivePackage"*), the low-level policy translates these discrete subtasks into continuous control actions like motion planning and force control. This architecture naturally extends to various robotic morphologies.
>
> Future extensions could incorporate temporal and state constraints on edges, or employ multi-level hierarchies to handle complex asynchronous planning and multi-agent coordination scenarios, further enhancing real-world applicability.
>
> > **W2**: It was a bit unclear how exactly low-level action policies are designed.
>
> Thank you for this comment. We have expanded **Sec 3.2&#x20;**&#x69;n the revised manuscript to provide a more detailed description of the low-level policy's design and implementation. We hope these clarifications address your question.
>
> > **W3**: Is task execution efficiency (*e.g*. #actions/#steps it took to reach a goal) a relevant metrics to consider perhaps?
>
> Thank you for this valuable suggestion. We would like to clarify that our primary contribution centers on planning enhancement, so we selected the task success rate (SR) and goal-conditioned success rate (GC) as the primary metrics to evaluate our framework's overall effectiveness. We agree that execution efficiency offers important complementary insights, although it is largely governed by the underlying navigation policy rather than our planning module. Following your suggestion, we have included an analysis of task execution efficiency in Appendix C.7. The results demonstrate that our method achieves competitive efficiency despite the replanning overhead.

---

> ### Author Response · Authors · 2025-11-23
>
> > **Q1:**  It is great that the scene graphs can be generated automatically on the fly. How truthful are scene graphs?&#x20;
>
> Our scene graph is specifically designed as a task-focused memory module for replanning, prioritizing the capture of task-relevant objects and relations over complete environmental reconstruction. This design choice naturally constrains the graph's scale and complexity, thereby improving accuracy. We further ensure representation reliability through multiple complementary mechanisms.
>
> First, we employ preventive measures through carefully designed few-shot prompts and **rule-based constraints**. For instance, we explicitly require that when predicates like "in" or "on" are detected, the "object" must be physically capable of holding the "subject." This prevents obvious relation errors during initial graph construction.
>
> Furthermore, the **dynamic update mechanism** continuously refines scene graph accuracy during execution. We employ pruning strategies, pose adjustments, and instance discrimination to handle repeated observations, diverse object types, and state transitions. Even with initially incomplete captures from certain perspectives, temporal integration of multi-view observations progressively enhances coverage of all task-relevant objects.
>
> The **LLM's inherent reasoning capability** provides additional robustness by leveraging commonsense understanding to compensate for occasional inconsistencies and minor perception errors.
>
> Collectively, these mechanisms enhance scene graph truthfulness while maintaining the advantages of automatic, on-the-fly construction. While perfect accuracy remains challenging, this reliability level has proven sufficient for robust replanning.
>
> > **Q2**: Why do most models perform better on the valid unseen set than the valid seen set in Figure 4?
>
> Thank you for this thoughtful observation. This performance difference stems from **the distribution of task complexity across splits**. As shown in Table S1, the valid-unseen split contains more simpler tasks (659 samples with 2-3 subtasks) compared to the valid-seen split (557 samples of the same length). Since shorter tasks are generally easier, this distributional difference explains the slightly higher success rates on the unseen set.
>
> Table S1: Distribution of task lengths and success rates across validation splits.
>
> | Length | Seen Samples | Accuracy on Seen | Unseen Samples | Accuracy on UnSeen |
> | ------ | ------------ | ---------------- | -------------- | ------------------ |
> | 2      | 236          | 97.46%           | 273            | 97.44%             |
> | 3      | 321          | 98.75%           | 386            | 97.67%             |
> | 5      | 118          | 90.68%           | 66             | 95.45%             |
> | 6      | 139          | 95.68%           | 81             | 95.06%             |
> | 8      | 6            | 33.33%           | 9              | 44.44%             |

---

> ### Author Response · Authors · 2025-11-26
> **Thank you for your timely reply!**
>
> We sincerely appreciate the time and effort you have invested in reviewing our submission and response. Your support and encouraging feedback are deeply motivating, and we are truly honored by your recommendation.

---

### Official Review · Reviewer_aMND · 2025-11-02

**Soundness:** 3
**Presentation:** 4
**Contribution:** 3
**Rating:** 8
**Confidence:** 3

**Summary:**

The paper introduces GraphPlan, a graph-enhanced planning framework for embodied agents that improves long-horizon reasoning and task execution in complex environments. It addresses key weaknesses of LLM planners such as hallucinated actions and poor generalization by integrating graph representations for reasoning. Through graph-guided prompting, verification, and RL, the system maintains grounded, feasible plans. Evaluated on the ALFRED benchmark, GraphPlan achieves SOTA performance, outperforming prior methods by about 6% in unseen success rate, and demonstrates strong generalization to long-horizon tasks.

**Strengths:**

- Symbolic structures for guiding LLM-based planning is novel and creative. The method is motivated intuitively.
- The dual graph method (task and scene decoupled) for provides a principled way to make LLMs operate for planning.
- The frameworks is well validated through comprehensive experiments, including ablations that isolate the contribution of each module.

**Weaknesses:**

- The construction of task graphs appears handcrafted/domain-dependent, raising concerns about how easily GraphPlan generalizes to new environments.
- In larger or more cluttered environments, these graphs could contain hundreds of nodes and relations. The paper assumes the LLM can interpret and reason over these graph descriptions accurately, but provides no analysis of performance degradation or prompt efficiency as graph complexity grows.
- Do the authors empirically demonstrate how GraphPlan’s performance scales with increasing task horizon length? While the benchmark shows overall results, there’s no detailed analysis of degradation trends of whether other methods fail progressively with more subtasks (longer horizons) while GraphPlan remains stable.

**Questions:**

- Could the authors provide additional evaluation of "reasoning fidelity". For example, how closely intermediate subgoals follow the intended logic of the instruction, or how often the model avoids invalid or redundant subtasks?
- How robust is GraphPlan when the scene graph contains missing or incorrect relations? Could the LLM detect and repair such inconsistencies through reasoning?
- While the long-horizon benchmark shows strong overall results, the paper doesn’t characterize computational or reasoning costs. How does inference time and #LLM calls scale with task length?

---

> ### Author Response · Authors · 2025-11-23
>
> > **W1**:  The construction of task graphs appears handcrafted/domain-dependent, raising concerns about how easily GraphPlan generalizes to new environments.
>
> We appreciate this concern and respectfully address the generalizability of our approach.
>
> **(i) Functional Compatibility-Based Graph Expansion:** While task graphs constructed solely from training data can only express tasks present in that data, our goal is to enable the task graph to serve as *universal representations within a given environment*. To achieve this, we automatically expand the graph by analyzing functional compatibility between subtasks. Specifically, we add directed edges between nodes when the terminal low-level action of a preceding subtask (*e.g., pickup*) is functionally compatible with the initiating action of a subsequent subtask (*e.g., put*) within the agent's capability range. This functional expansion mechanism enables both ***generalization to novel task combinations beyond training coverage and application to new tasks without expert demonstrations**.* Therefore, for new environments, we can design the task graph based on the environment's supported capabilities and their functional relationships.
>
> **(ii) Generalization Evidence:&#x20;**&#x57;e tested generalization to complex task compositions in **Sec. 4.3&#x20;**&#x61;nd achieved 90.04% success rate. To further demonstrate the task graph's scalability, we have incorporated experiments on novel skills in **Sec. 5** in our revised manuscript and GraphPlan maintains strong performance.
>
> **(iii) Applicability and Future Work:&#x20;**&#x4F;ur hierarchical planning framework applies naturally to domains with sequential dependencies and multi-solution paths. Future extensions could incorporate temporal and state constraints on edges, or employ multi-level hierarchies to handle complex asynchronous planning and multi-agent coordination scenarios.
>
> > **Q2**: The robustness to scene graph errors and LLM's ability to handle inconsistencies.
>
> Thank you for this comment. We acknowledge that VLM-based scene capture can be error-prone. To mitigate this, we employ rigorous strategies:
>
> (i) We implement preventive measures through *carefully designed few-shot prompts and rule-based constraints*. For instance, we explicitly require that when predicates like "in" or "on" are detected, the "object" must be physically capable of holding the "subject." This prevents obvious relation errors during initial graph construction.
>
> (ii) During execution, we account for repeated observations, object type variations, and state changes.  Our dynamic update mechanism maintains scene graph accuracy through *pruning strategies, pose updates, and instance differentiation during environmental interactions*. Even with incomplete captures from certain viewpoints, multi-view observations accumulate over time to ensure task-relevant objects are captured.
>
> (iii) The LLM's commonsense reasoning provides *error tolerance*. For example, if the VLM labels a bottle as "blue" but the task requires a "purple" bottle, the LLM infers that the blue bottle is the likely target when no other similar objects exist, preventing task failure due to minor perception errors.&#x20;
>
> > **W2**: Lack of analysis on performance with increasing graph complexity.
>
> Thank you for this insightful suggestion. We fully agree with your perspective and have added experiments to analyze the impact of task graph complexity. By expanding the original 12-node graph with 9 new nodes supported by AI2-Thor but unused in ALFRED, we have constructed extended task graphs with 16 and 21 nodes respectively. We evaluate performance of GraphPlan with task graph of different sizes, with success rates summarized below:
>
> Table S1: Performance comparison with different task graph complexity.
>
> | # Nodes | Valid Seen | Valid Unseen | Long Horizon |
> | ------- | ---------- | ------------ | ------------ |
> | 12      | 95.12      | 96.93        | 90.83        |
> | 16      | 94.88      | 95.71        | 86.68        |
> | 21      | 93.29      | 94.60        | 85.96        |
>
> The experimental results show only minor performance degradation even with increased graph complexity, demonstrating GraphPlan's strong scalability and robustness.

---

> ### Author Response · Authors · 2025-11-23
>
> > **W3**: Performance Degradation Analysis: Does the method's performance degrade progressively with longer task horizons compared to baselines?
> >
> > **Q1**: Reasoning Fidelity Evaluation: How faithfully does the model follow instruction logic and avoid invalid/redundant steps?
> >
> > **Q3**: Computational Cost Scaling: How do inference time and LLM call count scale with increasing task length?
>
> Thank you for these insightful suggestions. To address these points, we have conducted detailed experiments comparing GraphPlan against a strong baseline (Gemini-2.5-Pro with RAG) and the SFT with the identical backbone (Qwen2.5-7B-Instruct) across varying task horizons (7–12 steps). The results are shown in Table S2-S4, respectively.
>
> **(i) Performance Scaling**: GraphPlan shows minimal degradation with longer horizons, with accuracy declining from 98.73% to 80.71% while graph pass rates remain above 85%. This contrasts sharply with baselines: Gemini-2.5-Pro with RAG drops from 72.06% to 9.29%, and SFT collapses from 40.63% to near zero beyond 9 steps.
>
> **(ii) Reasoning Fidelity**: Following \[1], we employ four fine-grained error metrics to identify specific weaknesses in LLM planning. Quantitative analysis using fine-grained error metrics reveals GraphPlan's superior planning quality. It achieves remarkably low wrong transfer rates (0.02-2.16%) and near-zero affordance errors, demonstrating accurate action-object understanding through graph constraints. While missing and additional step rates show some increase (*e.g., additional steps: 1.14% to 8.81%*), they remain substantially lower than baselines (*e.g., Gemini's 10.81% to 66.67%*).
>
> **(iii) Computational Costs**: GraphPlan maintains stable inference times (\~7-9s) for 7-11 steps, demonstrating well-controlled overhead. Even at 12 steps, time only increases to 23.7s, significantly better than Gemini-2.5-Pro with RAG (210-336s). Despite verification mechanisms, LLM calls average only approximately 1.2, indicating most plans pass initial validation.
>
> These results collectively validate GraphPlan's robustness, efficiency, and reasoning quality across increasingly complex tasks.
>
> \[1] Li et al. "Embodied agent interface: Benchmarking llms for embodied decision making." NeurIPS, 2024
>
> Table S2:  Performance of GraphPlan with increasing task horizons.
>
> | Length | Avg. Accuracy (%) | Avg. Graph Pass Rate (%) | Avg. Missing Step Rate (%) | Avg. Additional Step Rate (%) | Avg. Wrong Transfer Rate (%) | Avg. Affordance Error Rate (%) | Avg. Inference Time (s) | Avg. LLM Calls |
> | ------ | ----------------- | ------------------------ | -------------------------- | ----------------------------- | ---------------------------- | ------------------------------ | ----------------------- | -------------- |
> | 7      | 98.73             | 99.68                    | 0.96                       | 1.14                          | 0.02                         | 0.12                           | 7.68                    | 1.05           |
> | 8      | 96.51             | 98.41                    | 1.12                       | 0.65                          | 0.06                         | 0.00                              | 7.82                    | 1.10            |
> | 9      | 85.04             | 96.33                    | 1.98                       | 1.44                          | 0.45                         | 0.00                              | 9.05                    | 1.21           |
> | 10     | 86.67             | 96.19                    | 0.54                       | 5.71                          | 0.26                         | 0.00                              | 9.34                    | 1.20            |
> | 11     | 81.43             | 95.71                    | 1.90                        | 8.81                          | 0.38                         | 0.00                              | 9.64                    | 1.21           |
> | 12     | 80.71             | 85.00                       | 5.90                        | 4.19                          | 2.16                         | 0.10                            | 23.70                    | 2.10            |

---

> ### Author Response · Authors · 2025-11-23
>
> Table S3: Performance of Gemini-2.5-Pro with RAG with increasing task horizons.
>
> | Length | Avg. Accuracy (%) | Avg. Graph Pass  Rate (%) | Avg. Missing Step Rate (%) | Avg. Additional Step Rate (%) | Avg. Wrong Transfer Rate (%) | Avg. Affordance Error Rate (%) | Avg. Inference Time (s) | Avg. LLM Calls |
> | ------ | ----------------- | ------------------------- | -------------------------- | ----------------------------- | ---------------------------- | ------------------------------ | ----------------------- | -------------- |
> | 7      | 72.06             | 83.17                     | 13.51                      | 10.81                         | 1.56                         | 0.15                           | 226.88                  | 1.00           |
> | 8      | 70.16             | 83.81                     | 12.38                      | 10.20                         | 1.90                         | 0.00                           | 210.78                  | 1.00           |
> | 9      | 29.40             | 62.47                     | 19.58                      | 10.83                         | 5.40                         | 0.40                           | 214.25                  | 1.00           |
> | 10     | 30.48             | 53.33                     | 15.00                      | 23.75                         | 4.51                         | 1.25                           | 221.46                  | 1.00           |
> | 11     | 15.71             | 80.00                     | 38.00                      | 22.45                         | 2.44                         | 0.00                           | 293.69                  | 1.00           |
> | 12     | 9.29              | 65.00                     | 38.33                      | 66.67                         | 0.91                         | 0.42                           | 335.96                  | 1.00           |
>
> Table S4: Performance of SFT with increasing task horizons.
>
> | Length | Avg. Accuracy (%) | Avg. Graph Pass  Rate (%) | Avg. Missing Step Rate (%) | Avg. Additional Step Rate (%) | Avg. Wrong Transfer Rate (%) | Avg. Affordance Error Rate (%) | Avg. Inference Time (s) | Avg. LLM Calls |
> | ------ | ----------------- | ------------------------- | -------------------------- | ----------------------------- | ---------------------------- | ------------------------------ | ----------------------- | -------------- |
> | 7      | 40.63             | 82.86                     | 29.92                      | 37.45                         | 2.21                         | 0.32                           | 7.65                    | 1.00           |
> | 8      | 41.90             | 81.27                     | 30.32                      | 29.65                         | 2.89                         | 0.10                           | 7.36                    | 1.00           |
> | 9      | 4.20              | 16.01                     | 26.76                      | 3.96                          | 11.20                        | 1.62                           | 7.16                    | 1.00           |
> | 10     | 5.71              | 20.00                     | 30.00                         | 39.82                         | 6.42                         | 0.14                           | 8.99                    | 1.00           |
> | 11     | 0.00              | 19.29                     | 28.10                       | 58.81                         | 6.36                         | 1.28                           | 9.38                    | 1.00           |
> | 12     | 3.57              | 6.43                      | 19.24                      | 0.00                          | 3.98                         | 6.18                           | 9.05                    | 1.00           |

---

> ### Author Response · Authors · 2025-11-27
>
> Dear Reviewer aMND,
>
> As the discussion period comes to a close, we would greatly appreciate it if you could kindly take a moment to review our responses at your convenience. If there are any remaining points that would benefit from further clarification, we would be more than happy to address them before the discussion window closes. Thank you again for your thoughtful review and for the time you've dedicated to our work.
>
> Sincerely,
>
> The Authors

---

### Official Review · Reviewer_MinK · 2025-11-02

**Soundness:** 2
**Presentation:** 2
**Contribution:** 2
**Rating:** 2
**Confidence:** 4

**Summary:**

The paper considers use of LLM’s for embodied agent planning, and proposes GraphPlan that uses a task graph and scene graph.  The task graph is developed a priori and characterizes the possible sequences of actions (sub-tasks).  This can be used in an LLM planning prompt to ensure the plan adheres to the graph.  A scene graph is constructed that incorporates the entire environment, with all relevant objects and relationships discretely encoded. The scene graph will be updated when changes occur.  The method is used on the ALFRED benchmark and some comparisons are made.

**Strengths:**

The methodology discretizes both the action space and the scene in graphs, and this provides a framework for LLM-based plan generation, replanning, and adhering to feasible plans. The planning method might be useful for highly characterized environments with a fixed set of simple actions and set of objects with relations encoded.

GRPO is used to generate a policy and rewards are linked to the graph structures. This guides the GRPO within the constraints specified for the tasks and scene.

The method is easy to understand and intuitively clear.

**Weaknesses:**

The method discretizes both action and scene, including objects and relationships.  This greatly simplifies the overall processing, which is logical but also highly constrained.  Given the task graph a priori, it isn't obvious that an LLM based planner is even needed, and why a graph searching type method can't be applied directly.

Scalability is unlikely.  The scene graph will grow with the scene size and will be a serious bottleneck, even in a highly controlled simple environment.  Adding new task and objects apparently requires a new policy training phase.

The overall novelty isn't obvious.  Learning scene graphs with objects has already been developed.  The verification of a finite LLM-plan has also been considered, e.g., by mapping to a finite automata type model.

The underlying task graph assumes each node can be reached (e.g., grasping), and this doesn't account for perception errors, incomplete sub-tasks, or other forms of real life conditions.

**Questions:**

How does the task graph approach compare with other LLM-based encoders that choose among a set of actions?

What is the novelty with respect to the scene graph?

---

> ### Author Response · Authors · 2025-11-23
>
> > **W1**: Why use an LLM planner instead of direct graph search given the predefined task graph?
>
> Thank you for the insightful question. **While graph search may appear to be a natural alternative, it faces fundamental limitations in high-level task planning.** First, the nodes in our task graph represent abstract task types defined by high-level actions and object meta-classes, not concrete subtasks. This abstraction **requires semantic grounding to map these meta-classes to instruction-specific objects before execution** by downstream low-level policies, while pure graph search lacks this capability. Furthermore, using graph search to explore possible paths still **requires a model to score candidate nodes** at each step, necessitating numerous LLM calls that incur high computational cost and create strong dependence on the scoring model's local accuracy.&#x20;
>
> To investigate this, we have conducted experiments using depth-first search with greedy/beam strategies, using LLMs for both candidate node scoring and meta-class to instruction-specific object mapping. Two model scales (i.e., Qwen-2.5-Instruct-7B and DeepSeek-R1-Distill-Qwen-32B) have been tested. Relevant discussions have been included in Appendix C.4, with experimental results as shown in Table S1 below.&#x20;
>
> Table S1: Comparisons with graph-search-based methods. &#x20;
>
> | Strategy      | LLM                          | Valid Seen | Valid Unseen | Long Horizon |
> | ------------- | ---------------------------- | ---------- | ------------ | ------------ |
> | Greedy Search | Qwen-2.5-Instruct-7B         | 19.02      | 19.39        | 2.58         |
> | Greedy Search | DeepSeek-R1-Distill-Qwen-32B | 46.34      | 49.08        | 12.82        |
> | Beam Search   | Qwen-2.5-Instruct-7B         | 17.93      | 18.16        | 1.86         |
> | Beam Search   | DeepSeek-R1-Distill-Qwen-32B | 40.70      | 43.68        | 11.39        |
> | GraphPlan     | Qwen-2.5-Instruct-7B         | 96.22      | 96.56        | 90.04        |
>
> The results show significantly worse performance, which we attribute to two factors: graph search makes locally optimal decisions at each step, easily leading to "myopic error accumulation" \[1] and suboptimal global paths, and the planning quality is limited by scoring accuracy. In contrast, GraphPlan enables holistic task reasoning and structural guidance through a synergistic combination of three mechanisms: (i) the task graph encoded in the instruction offers explicit planning guidance to the model, (ii) graph-augmented reinforcement learning helps the LLM acquire the ability to plan in reference to the task graph, and (iii) graph verification feedback improves the accuracy of initial plans.
>
> \[1] Yao et al. "Tree of thoughts: Deliberate problem solving with large language models." NeurIPS, 2023.

---

> ### Author Response · Authors · 2025-11-23
>
> > **Q1**: How does the task graph approach compare with other LLM-based encoders that choose among a set of actions?
>
> **We respectfully refer the reviewer to Sec. 4.3**, where we compared GraphPlan against multiple representative LLMs under diverse inference settings. GraphPlan achieves superior performance, particularly on complex long-horizon tasks with 90.04% success rate, substantially outperforming the best baseline (*i.e.*, 44.91%).  The newly added experiments on novel skills in **Sec. 5 validate the task graph's scalability**, demonstrating that the LLM can achieve zero-shot generalization to new tasks using the expanded graph. Beyond these empirical results, we would like to highlight three key advantages of our task graph approach:
>
> (i) LLM planners selecting from the action set are prone to hallucinations. The task graph enhances the planning capabilities of LLMs by modeling the logical constraints between subtasks. And this graph structure naturally accommodates multiple valid paths, making it particularly **suitable for multi-solution task planning problems**.
>
> (ii) The task graph serves as a universal representation that can be designed or expanded based on capabilities supported by the environment and their functional relationships (**Sec. 3.1.1**). This construction method enables coverage of novel task combinations beyond training coverage and application to new tasks without expert demonstrations (**Sec. 4.3&#x20;**&#x61;nd **Sec. 5**), while also showing promising few-shot learning potential (**Appendix C.8**).
>
> (iii) The task graph enables optimization and verification throughout the planning process. We incorporate task **graph constraint&#x20;**&#x69;n prompts during planning and replanning phases, while designing **edge-level and node-level rewards** for GRPO. The graph also serves as an **external verification tool** to provide immediate feedback and correction for LLM reasoning, substantially reducing planning hallucinations.
>
> These advantages collectively demonstrate GraphPlan's superiority over LLM-based action selection methods. Our hierarchical framework readily adapts to domains featuring sequential dependencies and multi-solution planning problems. Future work could extend this approach to complex asynchronous planning or multi-robot collaborative systems by **augmenting edges with constraints like task duration and state, or by constructing multi-layer graphs**.&#x20;

---

> ### Author Response · Authors · 2025-11-27
>
> > **W2 and Q2**: Concerns about scene graph scalability, policy retraining for new tasks/objects, and the novelty of the scene graph.
>
> Thank you for the question. We would like to clarify tha&#x74;**&#x20;our scene graph is scalable and adding new tasks or objects does not require policy retraining**. The scene graph is designed to provide task-specific environmental cues for the event-driven replanning module. It is generated dynamically from the agent’s egocentric views via a VLM during execution, without requiring preconstruction, and supports real-time updates.
>
> Compared to prior works, our scene graph construction method differs and improves in terms of task relevance and dynamic construction: &#x20;
>
> (i) General scene graphs \[2,3] contain numerous task-irrelevant objects and relations, which can overwhelm LLMs with excessive environmental details during planning. Our scene graph is specifically designed as a **task-focused memory module**, not a complete environment map. By storing only task-relevant objects, its size remains manageable (typically under 20 nodes in ALFRED, despite scenes containing around 68 object instances). This design naturally limits graph scale and complexity whil&#x65;**&#x20;directing planning attention to environmentally critical cues**, enabling more efficient and accurate reasoning. &#x20;
>
> (ii) Scene graphs are often generated statically \[3,4] and cannot adapt to object state changes or post-interaction environment updates. Some planning methods rely on pre-built scene graphs \[5,6], yet such ideally constructed graphs are often unavailable in real-world settings. Therefore, we adopt a **dynamic construction pipeline** based on the VLM and further ensure reliability through complementary mechanisms: &#x20;
>
> * To enhance VLM-based scene capture quality, we process both RGB and segmentation inputs to ensure label alignment and reduce ambiguity, while also implementing preventive measures through carefully designed few-shot prompts and rule-based constraints (*e.g.*, requiring physical plausibility for spatial relationships).
>
> * Our dynamic update mechanism maintains scene graph accuracy through *pruning strategies, pose updates, and instance differentiation during environmental interactions*. Even with incomplete captures from certain viewpoints, multi-view observations accumulate over time, thus ensuringcomprehensive coverage of task-relevant objects.
>
> \[2] Gu et al. "Conceptgraphs: Open-vocabulary 3d scene graphs for perception and planning." *ICRA*, 2024.
>
> \[3] Takmaz et al. "Search3d: Hierarchical open-vocabulary 3d segmentation." *IEEE Robotics and Automation Letters,&#x20;*&#x32;025.
>
> \[4] Wu et al. "Scenegraphfusion: Incremental 3d scene graph prediction from rgb-d sequences." *CVPR,* 2021.
>
> \[5] Rana et al. "Sayplan: Grounding large language models using 3d scene graphs for scalable robot task planning." *arXiv,&#x20;*&#x32;023.
>
> \[6]Liu et al. "Delta: Decomposed efficient long-term robot task planning using large language models."&#x20;*&#x20;ICRA*, 2025.
>
> > **W3**: The task graph's assumption of node reachability ignores real-world execution errors.&#x20;
>
> Thank you for your comments. We agree that assuming perfect node reachability is unrealistic, **which is precisely why we designed an event-driven replanning mechanism to address low-level execution uncertainties in** **Sec. 3.3**.&#x20;
>
> While the task graph provides structured knowledge for high-level planning, its open-loop nature lacks environmental awareness and self-correction. We therefore introduce an event-triggered replanning module driven by a dynamic scene graph to address low-level execution errors and realign plans with instruction intent. Additionally, our low-level policy incorporates interaction logging, object state tracking, and navigation error recovery to handle execution failures. For instance, when a  'Pickup(apple)' subtask fails in navigation, the agent first attempts recovery through environmental exploration. If the object remains unreachable after exhaustive search, the replanning module is triggered to propose alternative strategies, such as searching inside closed containers.
>
> **Effectiveness Evidence**: The effectiveness of this integrated approach is demonstrated by our sota result on the ALFRED benchmark, which contains realistic execution challenges like occluded objects and interaction failures.&#x20;

---

> ### Author Response · Authors · 2025-11-27
>
> Dear Reviewer MinK,
>
> As the discussion period comes to a close, we would greatly appreciate it if you could kindly take a moment to review our responses at your convenience. If there are any remaining points that would benefit from further clarification, we would be more than happy to address them before the discussion window closes. Thank you again for your thoughtful review and for the time you've dedicated to our work.
>
> Sincerely,
>
> The Authors

---

### Meta-Review · Area_Chair_1h7D · 2026-01-05

**Summary:**

While reviewers agreed that GraphPlan is a well-engineered system and achieves strong empirical performance on ALFRED, two reviewers raised fundamental concerns that inform the rejection decision. The primary objections focus on the core abstraction choices and scope of the approach: the reliance on a predefined task graph and discretized action abstractions raises questions about whether LLM-based planning is necessary compared to graph-search-based alternatives, and whether the proposed framework meaningfully advances beyond prior graph-based replanning and memory-augmented planning methods. Reviewers also expressed skepticism about scalability and generality, noting that the approach is demonstrated only in highly structured environments with curated action sets, and that scene graph complexity and task graph design may become bottlenecks in larger or less constrained settings. Although the rebuttal provided detailed clarifications and additional experiments—particularly addressing comparisons to graph search and execution uncertainty—these responses did not fully resolve concerns about novelty, abstraction validity, and applicability beyond benchmark-specific settings.

**Reviewer Concerns:**

Addressed concerns

- Comparison to graph search–based planning: The authors directly addressed the question of whether a predefined task graph makes LLM-based planning unnecessary by implementing greedy and beam-search baselines with LLM scoring. The results demonstrate significant performance degradation on long-horizon tasks, and the authors provide a plausible explanation based on myopic error accumulation and local scoring noise.

- Uncertainty and reachability assumptions: The rebuttal clarifies that the task graph is not assumed to be sufficient on its own, and that execution failures are handled through an event-driven replanning module informed by a dynamically updated scene graph. Additional explanations and examples illustrate how misaligned or failed subtasks trigger replanning.

- Empirical validation and ablations: The authors conducted extensive ablations isolating the contributions of the task graph, scene graph, reinforcement learning, and verification components, and provided controlled comparisons under identical backbones to demonstrate that performance gains are not due to increased model capacity or training data.

- Clarification of task graph construction: The authors explained how task graphs can be expanded using functional compatibility rather than relying solely on demonstrations, and provided additional experiments showing generalization to novel task compositions and skills without retraining.

Outstanding concerns:

- Scalability beyond structured benchmark environments: Despite arguments that task-centric scene graphs remain compact, the paper does not empirically demonstrate scalability to larger, more cluttered, or less structured environments. Concerns remain that both task graph complexity and scene graph maintenance could become bottlenecks outside ALFRED-scale settings.

- Dependence on predefined abstractions: The approach relies on predefined action abstractions, meta-classes, and task graphs tied to environment-specific capabilities. Reviewers questioned whether this limits applicability to new domains and undermines claims of general-purpose embodied planning.

- Novelty relative to prior work: Some reviewers questioned that the integration of task graphs, scene graphs, verification, and replanning constitutes a sufficiently novel conceptual advance beyond existing graph-based planning and memory-augmented methods, viewing the contribution primarily as system integration.

- Generality of conclusions: The strong empirical results are concentrated on a single benchmark, and it remains unclear whether the proposed framework would retain its advantages in environments with different perceptual challenges, action spaces, or task semantics.

**Reviewer Scores:**

- Reviewer MinK: No change (2). This reviewer’s objections are fundamental and abstraction-level, focusing on the necessity of LLM-based planning given a predefined task graph, scalability beyond structured environments, and overall novelty. While the rebuttal addressed specific technical questions (e.g., comparison to graph search and execution uncertainty), these responses do not address the fundamental concerns raised by the reviewer. A score increase is therefore unlikely.

- Reviewer ADQz: No change (2).T he reviewer raised concerns about novelty relative to prior graph-based replanning methods, meta-class design, and the generality of the proposed framework. Although the rebuttal provided clarifications and additional experiments, these concerns are primarily judgment-based rather than factual, making a score change unlikely.

- Reviewer aMND: No change (8) or lowering to 6. This reviewer was strongly positive and emphasized the novelty and completeness of the system-level contribution. The rebuttal reinforces this assessment but does not introduce fundamentally new evidence that would warrant a further increase. It might have been that the reviewer would even lower given the strong claims of the negative reviews.

-  Reviewer JYqU: No change (6).
This reviewer expressed a positive but measured assessment. The rebuttal addressed clarification requests but does not fundamentally change the scope or nature of the contribution. The reviewer reacted in the discussion and did not indicate a change of score.

---

### Decision · Program_Chairs · 2026-01-26

Reject